# ClimateLearn: Benchmarking Machine Learning for Weather and Climate Modeling

**Tung Nguyen**\*
UCLA
tungnd@cs.ucla.edu

**Jason Jewik**\*
UCLA
jason.jewik@ucla.edu

**Hritik Bansal**
UCLA
hbansal@ucla.edu

**Prakhar Sharma**
UCLA
prakhar6sharma@gmail.com

**Aditya Grover**
UCLA
adityag@cs.ucla.edu

## Abstract

Modeling weather and climate is an essential endeavor to understand the near- and long-term impacts of climate change, as well as inform technology and policymaking for adaptation and mitigation efforts. In recent years, there has been a surging interest in applying data-driven methods based on machine learning for solving core problems such as weather forecasting and climate downscaling. Despite promising results, much of this progress has been impaired due to the lack of large-scale, open-source efforts for reproducibility, resulting in the use of inconsistent or underspecified datasets, training setups, and evaluations by both domain scientists and artificial intelligence researchers. We introduce `ClimateLearn`, an open-source PyTorch library that vastly simplifies the training and evaluation of machine learning models for data-driven climate science. `ClimateLearn` consists of holistic pipelines for dataset processing (e.g., ERA5, CMIP6, PRISM), implementation of state-of-the-art deep learning models (e.g., Transformers, ResNets), and quantitative and qualitative evaluation for standard weather and climate modeling tasks. We supplement these functionalities with extensive documentation, contribution guides, and quickstart tutorials to expand access and promote community growth. We have also performed comprehensive forecasting and downscaling experiments to showcase the capabilities and key features of our library. To our knowledge, `ClimateLearn` is the first large-scale, open-source effort for bridging research in weather and climate modeling with modern machine learning systems. Our library is available publicly at `https://github.com/aditya-grover/climate-learn`.

## 1 Introduction

The escalating extent, duration, and severity of extreme weather events such as droughts, floods, and heatwaves in recent decades are some of the most devastating outcomes of climate change. Moreover, as average surface temperature is anticipated to continue rising through the end of the century, such extreme weather events are likely to occur with even greater intensity and frequency in the future [12, 33, 43, 46, 67, 74]. The key devices used by scientists to understand historical trends and make such predictions about future weather and climate are the numerical weather prediction (NWP) models. These models represent Earth system components including the atmosphere, land surface, ocean, and sea ice as intricate dynamical systems, and they are the prevailing paradigm for weather and climate modeling today due to their established reliability, well-founded design, and extensive

---

\*Equal contribution.

37th Conference on Neural Information Processing Systems (NeurIPS 2023) Track on Datasets and Benchmarks.

study [3, 13, 29, 51]. However, they also suffer from notable limitations such as inadequate resolution of several subgrid processes, coarse representation of local geographical features, incapacity to utilize sources of observational data (e.g., weather stations, radar, satellites) in an automated manner, and substantial demand for computing resources [2, 39, 40, 64]. These deficiencies combined with the expanding availability of petabyte-scale climate data [18, 19, 50] and lowering compute requirements of machine learning (ML) models in recent years have motivated researchers from both the climate science and artificial intelligence (AI) communities to investigate the application of ML-based methods in weather and climate modeling [7, 14, 35, 36, 44, 47, 59, 77, 89].

In spite of this growing interest, the improvements have been marred by the lack of practically grounded data benchmarks, open-source model implementations, and transparency in evaluation. For example, many papers in weather forecasting [4, 8, 17, 22, 37, 54, 65, 75, 80, 86, 87, 90], climate projection [84], and climate downscaling [1, 41, 49, 68, 70, 82] choose to benchmark on different geographical regions, temporal ranges, evaluation metrics, and data augmentation strategies. These inconsistencies can confound the source of reported improvements and promotes a culture of irreproducible scientific practices [34]. Recently, there have been some leaderboard benchmarks, such as WeatherBench [63], ClimateBench [84], and FloodNet [61], that propose datasets and baselines for specific tasks in climate science, but a holistic software ecosystem that encompasses the entire data, modeling, and evaluation pipeline across several tasks is lacking.

To bridge this gap, we propose `ClimateLearn`, an open-source, user-friendly PyTorch library for data-driven climate science. To the best of our knowledge, it is the first software package to provide end-to-end ML pipelines for weather and climate modeling. `ClimateLearn` supports data pre-processing utilities, implements popular deep learning models along with traditional baseline methods, and enables easy quantification and visualization of data and model predictions for fundamental tasks in climate science, including weather forecasting, downscaling, and climate projections. One segment of `ClimateLearn`'s target user demographic are weather and climate scientists, who possess expertise in the physical laws and phenomena relevant for constructing robust modeling priors, but might lack familiarity with optimal approaches for implementing, training, and evaluating machine learning models. Another segment of `ClimateLearn`'s target user demographic are ML researchers, who might encounter difficulties framing weather and climate modeling problems in a scientifically sound and practically useful manner, working with climate datasets—which is quite heterogeneous and often exists in bespoke file formats uncommon in mainstream ML research (e.g., NetCDF), or appropriately quantifying and visualizing their results for interpretation and deployments.

To showcase the capabilities of `ClimateLearn` and establish benchmarks, we perform and report results from numerous experiments on the supported tasks with a variety of traditional methods and our own tuned implementations of deep learning models on weather and climate datasets. In addition to traditional evaluation setups, we have created novel dataset and benchmarking scenarios to test model robustness and applicability to forecasting extreme weather events. Further, the library is modular and easily extendable to include additional tasks, datasets, models, metrics, and visualizations. We have also provided extensive documentation and contribution guides for improving the ease of community adoption and accelerating open-source expansion. While our library is already public, we are releasing all our data, code, and model checkpoints for the benchmarked evaluation in this paper to aid reproducibility and broader interdisciplinary research efforts.

## 2 Related work

Recent works have proposed benchmark datasets for weather and climate modeling problems. Prominently, Rasp et al. [63] proposed WeatherBench, a dataset for weather forecasting based on ERA5, followed by an extension called WeatherBench Probability [21], which adds support for probabilistic forecasting. Mouatadid et al. [48] extend similar benchmarks to the subseasonal to seasonal timescale. For precipitation events such as rain specifically, there are prior datasets such as RainBench [15] and IowaRain [73]. There exist datasets such as ExtremeWeather [60], FloodNet [61], EarthNet [66], DroughtED [45] and ClimateNet [56] for detection and localization of extreme weather events, and NADBenchmarks [58] for natural disasters related tasks. Cachay et al. [9] recently proposed ClimART, a benchmark dataset for emulating atmospheric radiative transfer in weather and climate models. For identifying long-term, globally-averaged trends in climate, Watson-Parris et al. [84] proposed ClimateBench, a dataset for climate model emulation.

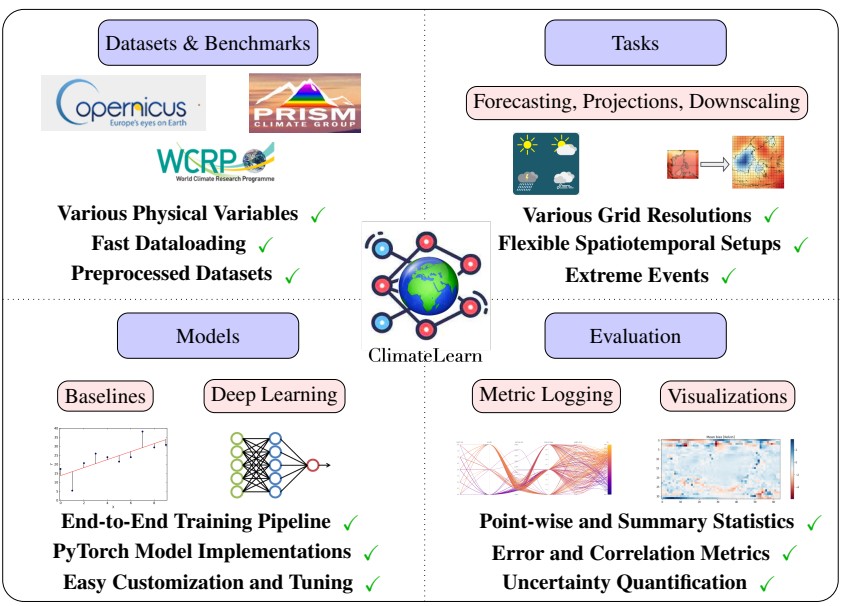

Figure 1: Key components of `ClimateLearn`. We support observational, simulated, and reanalysis datasets from a variety of sources. The currently supported tasks are weather forecasting, downscaling, and climate projection. `ClimateLearn` also provides a suite of standard baselines and deep learning architectures, along with common metrics, visualizations, and logging support.

CliMetLab [11] is a library which aims to simplify the process of downloading, preprocessing, and making climate data ML-friendly. While CliMetLab focuses purely on providing easy access to climate data, `ClimateLearn` is an end-to-end library for training and evaluating machine learning models on weather and climate problems. Moreover, while CliMetLab is intended to be used in Jupyter notebooks, `ClimateLearn` encompasses a wider range of usecases, from running quick starter code in Jupyter to large-scale benchmarking experiments.

Beyond plain datasets, libraries such as Scikit-downscale [24], CCdownscaling [55], and CMIP6-Downscaling [10] provide tools for post-processing of climate model outputs via statistical, non-deep-learning downscaling, or mapping low-resolution gridded, image-like inputs to high-resolution gridded outputs. In a slightly different approach, pyESD focuses on downscaling from gridded climate data to specific weather stations [6]. Pyrocast [79] proposes an integrated ML pipeline to forecast Pyrocumulonimbus (PyroCb) Clouds. Many of these works supply only individual components of an ML pipeline but do not always have an API for loading climate data into a ML-ready format, or standard model implementations and evaluation protocols across multiple climate science tasks. As an end-to-end ML pipeline, `ClimateLearn` holistically bridges the gap for applying ML to challenging weather and climate modeling tasks like forecasting, downscaling, and climate projection.

## 3 Key Components of `ClimateLearn`

`ClimateLearn` is a PyTorch library that implements a range of functionalities for benchmarking of ML models for weather and climate. Broadly, our library is comprised of four components: tasks, datasets, models, and evaluations. See Figure 1 for an illustration. Sample code snippets for configuring each component is provided in Appendix E.

### 3.1 Tasks

**Weather forecasting** is the task of predicting the weather at a future time step $t + \Delta t$ given the weather conditions at the current step $t$ and optionally steps preceding $t$. A ML model receives an input of shape $C \times H \times W$ and predicts an output of shape $C' \times H \times W$. $C$ and $C'$ denote the number of input and output channels, respectively, which contain variables such as geopotential, temperature, and humidity. $H$ and $W$ denote the spatial coverage and resolution of each channel, which depend

on the region studied and how densely we grid it. In our benchmarking, we focus on forecasting all gird points globally, but `ClimateLearn` can be easily extended to regional forecasting.

**Downscaling** Due to their high computational cost, existing climate models often use large grid cells, leading to low-resolution predictions. While useful for understanding large-scale climate trends, these do not provide sufficient detail to analyze local phenomena and design regional policies. The process of correcting biases in climate model outputs and mapping them to higher resolutions is known as downscaling. ML models for downscaling are trained to map an input of shape $C \times H \times W$ to a higher resolution output $C' \times H' \times W'$, where $H' > H$ and $W' > W$. As in forecasting, in downscaling, $H \times W$ and $H' \times W'$ can span either the entire globe or a specific region.

**Climate projection** aims to obtain long-term predictions of the climate under different forcings, e.g., greenhouse gas emissions. We provide support to download data from ClimateBench [84], a recent benchmark designed for testing ML models for climate projections. Here, the task is to predict the annual mean distributions of 4 climate variables: surface temperature, diurnal temperature range, precipitation, and the 90th percentile of precipitation, given four anthropogenic forcing factors: carbon dioxide ($CO_2$), sulfur dioxide ($SO_2$), black carbon (BC), and methane ($CH_4$).

### 3.2 Datasets

**ERA5** is a commonly-used data source for training and benchmarking data-driven forecasting and downscaling methods [4, 38, 49, 52, 54, 63, 62]. It is maintained by the European Center for Medium-Range Weather Forecasting (ECMWF) [29]. ERA5 is a reanalysis dataset that provides the best guess of the state of the atmosphere and land-surface variables at any point in time by combining multiple sources of observational data with the forecasts of the current state-of-the-art forecasting model known as the Integrated Forecasting System (IFS) [85]. In its raw format, ERA5 contains hourly data from 1979 to the current time on a $0.25°$ grid of the Earth's sphere, with different climate variables at 37 different pressure levels plus the Earth's surface. This corresponds to nearly 400,000 data points with a resolution of $721 \times 1440$. As this data is too big for most deep learning models, `ClimateLearn` also supports downloading a smaller version of ERA5 from WeatherBench [63], which uses a subset of ERA5 climate variables and regrids the raw data to lower resolutions.

**Extreme-ERA5** is a subset of ERA5 that we have constructed to evaluate forecasting performance in extreme weather situations. Specifically, we consider "simple extreme" events [83, 5], i.e., weather events that have individual climate variables exceeding critical values locally. Heat waves and cold spells are examples of extreme events that can be quantitatively captured by extreme localized surface-level temperatures over prolonged days. To mimic real-world scenarios, we calculate thresholds for each pixel of the grid using the 5th and 95th percentile of the 7-day localized mean surface temperature over the training period (1979-2015). We then select a subset of pixels from all the available pixels in the testing set (2017-18) that had a 7-day localized mean surface temperature beyond these thresholds. We refer to Appendix B.2.2 for more details.

**CMIP6** is a collection of simulated data from the Coupled Model Intercomparison Project Phase 6 (CMIP6) [19], an international effort across different climate modeling groups to compare and evaluate their global climate models. While the main goal of CMIP6 is to improve the understanding of Earth's climate systems, the data from their experimental runs is freely accessible online. CMIP6 data covers a wide range of climate variables, including temperature and precipitation, from hundreds of climate models, providing a rich source of data. For our forecasting experiments, we specifically use the ouputs of CMIP6's MPI-ESM1.2-HR model, as it contains similar climate variables to those represented in ERA5 and was also considered in previous works for pretraining deep learning models [62]. MPI-ESM1.2-HR provides data from 1850 to 2015 with a temporal resolution of 6 hours and spatial resolution of $1°$. Since this again corresponds to a grid that is too big for most deep learning models, we provide lower resolution versions of this dataset for training and evaluation. Besides, we also perform experiments with ClimateBench, which contains data on a range of future emissions scenarios based on simulations by the Norwegian Earth System Model [71], another member of CMIP6. We refer to Appendix B.2.3 for time ranges and more details of the experiments.

**PRISM** is a dataset of various observed atmospheric variables like precipitation and temperature over the conterminous United States at varying spatial and temporal resolutions from 1895 to present day. It is maintained by the PRISM Climate Group at Oregon State University [57]. At the highest publicly available resolution, PRISM contains daily data on a grid of 4 km by 4 km cells (approximately

$0.03°$), which corresponds to a matrix of shape $621 \times 1405$. For the same reason we regrid ERA5 and CMIP6, we also provide a regridded version of raw PRISM data to $0.75°$ resolution.

### 3.3 Models

**Traditional baselines** `ClimateLearn` provides the following traditional baseline methods for forecasting: climatology, persistence, and linear regression. The climatology method uses historical average values of the predictands as the forecast. In `ClimateLearn`, we consider two versions of the climatology, one in which we compute the average value over the entire training set, and the other keeps a mean for each of the 52 calendar weeks to account for the seasonal cycle of the climate. The persistence method uses the last observed values of the predictands as the forecast. For downscaling, `ClimateLearn` provides nearest and bilinear interpolation. Nearest interpolation estimates the value of an unknown pixel to be the value of the nearest known pixel. Bilinear interpolation estimates the value at an unknown pixel by taking the weighted average of neighboring pixels.

**Deep learning models** The data for gridded weather and climate variables is represented as a 3D matrix, where latitude, longitude, and the variables form the height, width, and channels, respectively. Hence, convolutional neural networks (CNNs) are commonly used for forecasting and downscaling, which can be viewed as instances of the image-to-image translation problem [17, 30, 49, 62, 68, 72, 75, 80, 81, 86, 87]. `ClimateLearn` supports ResNet [27] and U-Net [69]—two prominent variants of the commonly used CNN architectures. Additionally, `ClimateLearn` supports Vision Transformer (ViT) [4, 20, 52], a class of models that represent images as a sequence of pixel patches. `ClimateLearn` also supports loading benchmark models from the literature such as Rasp and Thuerey [62] in a single line of code and is built so that custom models can be added easily.

### 3.4 Evaluations

**Forecasting metrics** For deterministic forecasting, `ClimateLearn` provides metrics such as root mean square error (RMSE) and anomaly correlation coefficient (ACC), which measures how well model forecasts match ground truth anomalies. For probabilistic forecasting, `ClimateLearn` provides spread-skill ratio and continuous ranked probability score, as defined by Garg et al. [21]. `ClimateLearn` also provides latitude-weighted version of these metrics, which lends extra weight to pixels near the equator. This is needed because the curvature of the Earth means that grid cells at low latitudes cover less area than grid cells at high latitudes. We refer to Appendix B.4 for additional details, including equations.

**Downscaling metrics** For downscaling, `ClimateLearn` uses RMSE, mean bias, and Pearson's correlation coefficient, in which mean bias is the difference between the spatial mean of ground-truth values and the spatial mean of predictions. We refer to Appendix B.4 for additional details.

**Climate projection metrics** In addition to the standard RMSE metric, we provide two metrics suggested by ClimateBench: Normalized spatial root mean square error ($\text{NRMSE}_s$) and Normalized global root mean square error ($\text{NRMSE}_g$). We refer to Appendix B.4 for more details.

**Visualization** Besides these quantitative evaluation procedures, `ClimateLearn` also provides ways for users to inspect model performance qualitatively through visualizations of data and model predictions. For instance, in a single line of code, users can visually inspect their forecasting model's per-pixel mean bias, or the expected values of forecast errors, over the testing period. Such a visualization can be useful for pinpointing the regions on which the model's predictions consistently deviate from the ground truth in a certain direction. For probabilistic forecasts, `ClimateLearn` can generate the corresponding rank histogram, which indicates the reliability and sharpness of the model. Sample visualizations of deterministic and probabilistic predictions are provided in Appendix D.

## 4 Benchmark Evaluation via `ClimateLearn`

In this section, we evaluate the performance of different deep learning methods supported by `ClimateLearn` on weather forecasting and climate downscaling. We refer to Appendix C.1 for experiments on the climate projection task. We conduct extensive experiments and analyses with different settings to showcase the features and flexibility of our library.

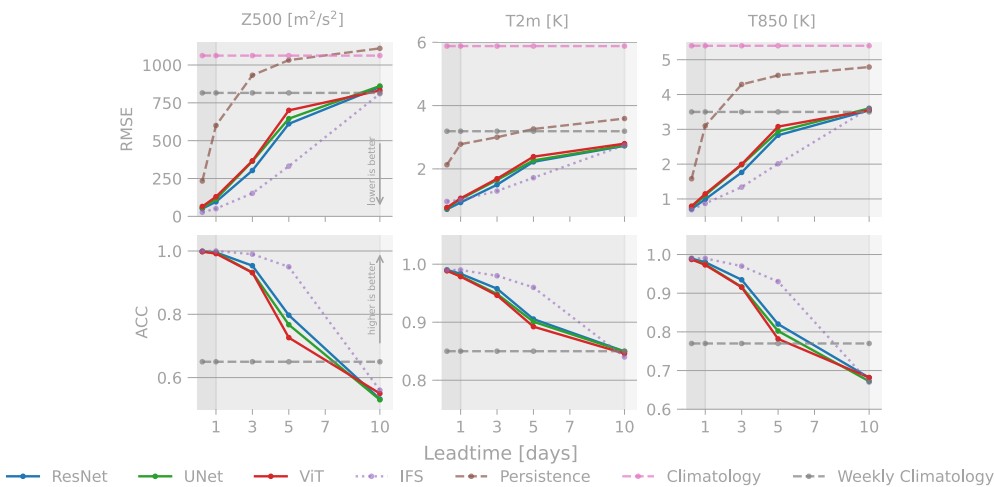

Figure 2: Performance on forecasting three variables at different lead times. Solid lines are deep learning methods, dashed lines are simple baselines, and the dotted line is the physics-based model. Lower RMSE and higher ACC indicate better performance.

## 4.1 Weather forecasting

We first benchmark on weather forecasting. In addition, we compare different approaches for training forecast models in Section 4.1.1, and investigate the robustness of these models to extreme weather events and data distribution shift in Section 4.1.2 and 4.1.3, respectively.

**Task** We consider the task of forecasting the geopotential at 500hPa (Z500), temperature at 850hPa (T850), and temperature at 2 meters from the ground (T2m) at five different lead times: 6 hours, and $\{1, 3, 5, 10\}$ days. Z500 and T850 are often used for benchmarking in previous works [4, 38, 52, 54, 62, 63], while the surface variable T2m is relevant to human activities.

**Baselines** We consider ResNet [27], U-Net [69], and ViT [16] which are three common deep learning architectures in computer vision. We provide the architectural details of these networks in Appendix B.1. We perform direct forecasting, where we train one neural network for each lead time. In addition, we compare the deep learning methods with climatology, persistence, and IFS [85].

**Data** We use ERA5 [29] at $5.625°$ for training and evaluation, which is equivalent to having a $32 \times 64$ grid for each climate variable. The input variables to the deep learning models include geopotential, temperature, zonal and meridional wind, relative humidity, and specific humidity at 7 pressure levels $(50, 250, 500, 600, 700, 850, 925)$hPa, 2-meter temperature, 10-meter zonal and meridional wind, incoming solar radiation, and finally 3 constant fields: the land-sea mask, orography, and the latitude, which together constitute 49 input variables. For non-constant variables, we use data at 3 timesteps $t$, $t - 6h$, and $t - 12h$ to predict the weather at $t + \Delta t$, resulting in $46 \times 3 + 3 = 141$ input channels. Each channel is standardized to have a mean of 0 and a standard deviation of 1. The training period is from 1979 to 2015, validation in 2016, and test in 2017 and 2018.

**Training and evaluation** We use latitude-weighted mean squared error as the loss function. We use AdamW optimizer [42] with a learning rate of $5 \times 10^{-4}$ and weight decay of $1 \times 10^{-5}$, a linear warmup schedule for 5 epochs, followed by cosine-annealing for 45 epochs. We train for 50 epochs with 128 batch size, and use early stopping with a patience of 5 epochs. We use latitude-weighted root mean squared error (RMSE) and anomaly correlation coefficient (ACC) as the test metrics.

**Benchmark results** Figure 2 shows the performance of different baselines. As expected, the forecast quality in terms of both RMSE and ACC of all baselines worsens with increasing lead times. The deep learning methods significantly outperform climatology and persistence but underperform IFS. ResNet is the best-performing deep learning model on most tasks in both metrics. We hypothesize that while being more powerful than ResNet in general, U-Net tends to perform better when trained on high-resolution data [69], and ViT often suffers from overfitting when trained from scratch [28, 52]. Our reported performance of ResNet closely matches that of previous work [62].

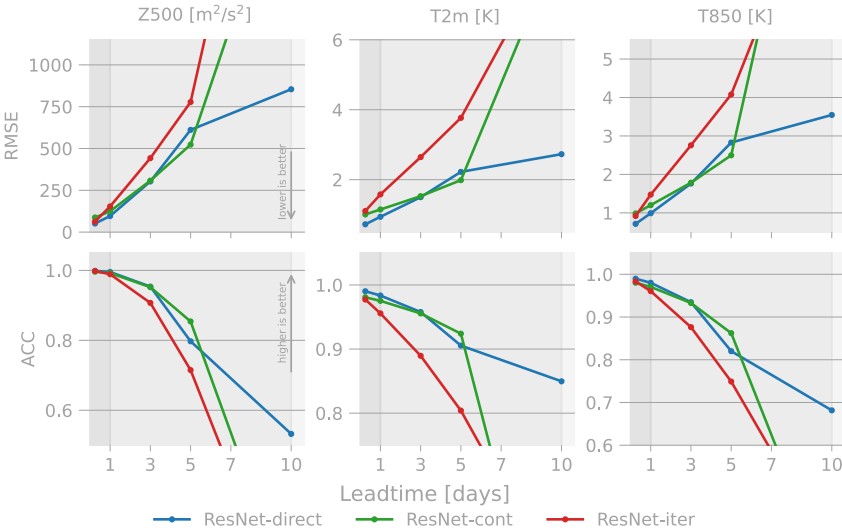

Figure 3: Comparison of direct, continuous, and iterative forecasting with ResNet architecture.

#### 4.1.1 Should we perform direct, continuous, or iterative forecasting?

In direct forecasting, we train a separate model for each lead time. This can be computationally expensive as the training cost scales linearly with the number of lead times. In this section, we consider two alternative approaches, namely, continuous forecasting and iterative forecasting, and investigate the trade-off between computation and performance. In continuous forecasting, a model conditions on lead time information to make corresponding predictions, which allows the same trained model to make forecasts at any lead times. We refer to Appendix B.3.1 for details on how to do this. In iterative forecasting, we train the model to forecast at a short lead time, i.e., 6 hours, and roll out the predictions during evaluation to make forecasts at longer horizons. We note that in order to roll out more than one step, the model must predict all variables in the input. This provides the benefit of training a single model that can predict at any lead time that is a multiplication of 6.

We compare the performance of direct, continuous, and iterative forecasting using the same ResNet architecture with training and evaluation settings identical to Section 4.1. Figure 3 shows that ResNet-cont slightly underperforms the direct model at 6-hour and 1-day lead times, but performs similarly or even better at 3-day and 5-day forecasting. We hypothesize that for difficult tasks, training with randomized lead times enlarges the training data and thus improves the generalization of the model. A similar result was observed by Rasp et al. [63]. However, the continuous model does not generalize well to unseen lead times, which explains the poor performance when evaluated at 10-day forecasting. ResNet-iter performs the worst in the three approaches, which achieves a reasonable performance at 6-hour lead time, but the prediction error accumulates exponentially at longer horizons. This was also observed in previous works [52, 63]. We believe this issue can be mitigated by multi-step training [54], which we leave as future work.

#### 4.1.2 Extreme weather prediction

Despite the surface-level temperature being an outlier, Table 1 shows that deep learning and persistence perform better on Extreme-ERA5 than ERA5 for all different lead times. Climatology performs worse on Extreme-ERA5, which is expected since predicting the mean of the target distribution is not a good strategy for outlier data. The persistence performance indicates that the variation between the input and output values is comparatively less for extreme weather conditions.

Table 1: Latitude-weighted RMSE on the normal and extreme test splits of ERA5.

| T2M | 6 Hours | 1 Day | 3 Days |
|---|---|---|---|
| Climatology | 5.87 / 6.51 | 5.87 / 6.53 | 5.87 / 6.58 |
| Persistence | 2.76 / 2.99 | 2.13 / 1.78 | 2.99 / 2.42 |
| ResNet | 0.72 / **0.72** | 0.94 / **0.91** | 1.50 / **1.33** |
| U-Net | 0.76 / 0.77 | 1.04 / 0.99 | 1.65 / 1.43 |
| ViT | 0.78 / 0.80 | 1.09 / 1.05 | 1.71 / 1.55 |

conditions. We hypothesize that, although the marginal distribution $p(y)$ of the subset data is extreme, the conditional distribution $p(y|x)$ which we are trying to model is not extreme. Thus, we are not

Table 3: Downscaling experiments on ERA5 (5.6°) to ERA5 (2.8°) and ERA5 (2.8°) to PRISM (0.75°). For ERA5 (5.6°) to ERA5 (2.8°), Pearson's correlation coefficient was 1.0 for all models.

| | ERA5 to ERA5 | | | | | | ERA5 to PRISM | | |
| | Z500 (m² s⁻²) | | T850 (K) | | T2m (K) | | Daily Max T2m (K) | | |
| | RMSE | Mean bias | RMSE | Mean bias | RMSE | Mean bias | RMSE | Mean bias | Pearson |
|---|---|---|---|---|---|---|---|---|---|
| Nearest | 269.67 | **0.04** | 1.99 | **0.00** | 3.11 | **0.00** | 2.91 | **-0.05** | 0.89 |
| Bilinear | 134.07 | **0.04** | 1.50 | **0.00** | 2.46 | **0.00** | 2.64 | 0.12 | 0.91 |
| ResNet | 54.20 | −6.41 | **0.39** | −0.05 | **1.10** | −0.22 | 1.86 | −0.11 | 0.95 |
| Unet | **43.84** | −6.55 | 0.94 | −0.06 | **1.10** | −0.12 | **1.57** | −0.14 | **0.97** |
| ViT | 85.32 | −35.98 | 1.03 | −0.01 | 1.25 | −0.20 | 2.18 | −0.26 | 0.94 |

experiencing any drop in performance on such a subset. While from a ML standpoint, it might seem necessary to evaluate the models for cases where the conditional distribution is extreme for any input variable, such a dataset might not qualify under the well-known categories of extreme events. Future studies for constructing extreme datasets could try targeting extreme events such as floods, which are usually caused by high amounts of precipitation under a short period of time [76].

### 4.1.3 Data robustness of deep learning models

We study the impact of data distribution shifts on forecasting performance. We consider CMIP6 and ERA5 as two different data sources. The input variables are similar to the standard setting, except that we remove relative humidity, 10-meter zonal and meridional wind, incoming solar radiation, and the 3 constant fields due to their unavailability in CMIP6. To account for differences in the temporal resolution and data coverage, we set the temporal resolution to 6 hours and set 1979-2010, 2011-12, and 2013-14 as training, validation, and testing years respectively.

Table 2: Performance of ResNet trained on one dataset (columns) and evaluated on another (rows).

| | | ERA5 | | CMIP6 | |
| | | ACC | RMSE | ACC | RMSE |
|---|---|---|---|---|---|
| | Z500 | **0.95** | **322.86** | 0.93 | 345.00 |
| ERA5 | T850 | **0.93** | **1.90** | 0.90 | 2.21 |
| | T2m | **0.95** | **1.62** | 0.93 | 1.94 |
| | Z500 | 0.95 | 357.66 | **0.96** | **306.86** |
| CMIP6 | T850 | 0.91 | 2.11 | **0.94** | **1.70** |
| | T2m | 0.93 | 1.91 | **0.96** | **1.53** |

Table 2 shows that all methods achieve better evaluation scores if the training and testing splits come from the same dataset, but cross-dataset performance is not far behind, highlighting the robustness of the models across distributional shifts. We see a similar trend for different models across different lead times, which we refer to Appendix C.3 for more details. We also conducted an experiment where the years 1850-1978 are included in training for CMIP6. The results show that for all models across almost all lead times, training on CMIP6 leads to even better performance on ERA5 than training on ERA5. For exact numbers and setup refer to Appendix C.3.

## 4.2 Downscaling

**Task and data** We consider two settings for downscaling. In the first setting, we downscale 5.625° ERA5 data to 2.8125° ERA5, both at a global scale and hourly intervals. The input and target variables are the same as used in Section 4.1. In the second setting, we consider downscaling 2.8125° ERA5 data over the conterminous United States to 0.75° PRISM data over the same region at daily intervals. This is equivalent to downscaling a reanalysis/simulated dataset to an observational dataset, similar to previous papers [25, 68, 78]. The cropped ERA5 data has shape $9 \times 21$ while the regridded PRISM data is padded with zeros to the shape $32 \times 64$. The only input and output variable is daily max T2m, which is normalized to have 0 mean and 1 standard deviation. The training period is from 1981 to 2015, validation is in 2016, and the testing period is from 2017 to 2018.

**Baselines** We compare ResNet, U-Net, and ViT with two baselines: nearest and bilinear interpolation.

**Training and evaluation** We use MSE as the loss function with the same optimizer and learning rate scheduler as in Section 4.1, with an initial learning rate of $1 \times 10^{-5}$. A separate model is trained for each output variable, and all models post-process the results of bilinear interpolation. We use RMSE, Pearson's correlation coefficient, and mean bias as the test metrics. All metrics are masked properly since PRISM does not have data over the oceans. See Appendix B.4 for further details.

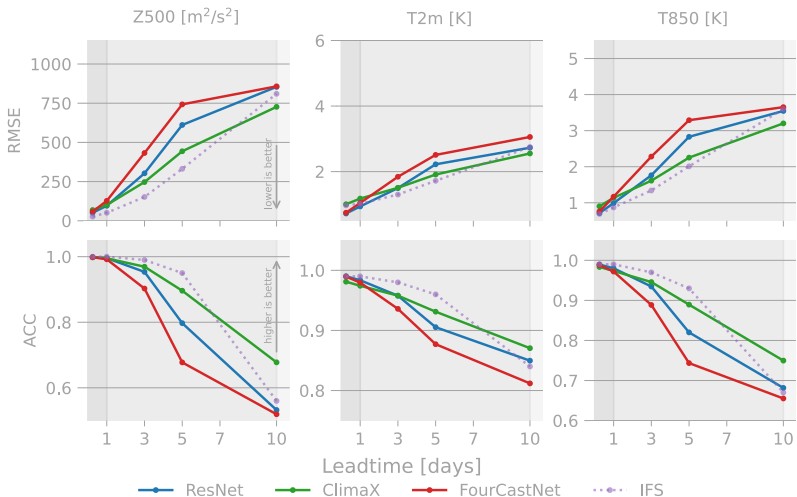

Figure 4: Forecasting performance of recent SoTA deep learning models.

Table 4: Performance of the two baselines in probabilistic weather forecasting.

| | RMSE of ensemble mean (5/10 days) | | | ACC of ensemble mean (5/10 days) | | | CRPS (5/10 days) | | | Spread-Skill Ratio (5/10 days) | | |
|---|---|---|---|---|---|---|---|---|---|---|---|---|
| | Z500 | T850 | T2m | Z500 | T850 | T2m | Z500 | T850 | T2m | Z500 | T850 | T2m |
| ResNet-Parametric | 610/856 | 2.83/3.61 | 2.21/2.74 | 0.80/0.52 | 0.82/0.67 | 0.91/0.85 | 344.4/488.3 | 1.77/2.30 | 1.24/**1.54** | 0.27/0.35 | 0.32/0.39 | 0.36/0.39 |
| ResNet-Ensemble | **572/807** | **2.67/3.47** | **2.09**/2.74 | **0.82/0.57** | **0.84/0.69** | **0.92**/0.85 | **323.9/474.8** | **1.68**/2.30 | **1.19**/1.61 | 0.38/0.29 | 0.34/0.26 | 0.34/0.37 |

**Benchmark results** Table 3 shows the performance of different baselines in both settings. As expected for the first setting, all methods achieve relatively low errors. The deep learning models outperformed both interpolation methods significantly on RMSE, but tend to overestimate the target variables, leading to negative mean bias. In the second setting—where the input and output come from two different datasets—the performance of all baselines drops. Nonetheless, the deep learning models again outperform the baseline methods on RMSE and exhibit negative mean bias, but also achieve higher Pearson's correlation coefficients.

## 5 Additional experiments

### 5.1 Benchmarking recent SoTA methods

We develop `ClimateLearn` as a long-term sustainable project that allows easy extension to new models, tasks, and datasets by us and others (via open-source). To demonstrate this, we added ClimaX [52] and FourCastNet [54] as two state-of-the-art deep learning models for weather forecasting to `ClimateLearn`. Figure 4 benchmarks these two methods against ResNet and IFS. While ClimaX follows IFS closely and even surpasses this strong baseline at the 10-day lead time, FourCastNet performs poorly on low-resolution data.

### 5.2 Probabilistic weather forecasting

To further showcase the flexibility and extensibility of `ClimateLearn`, we added probabilistic weather forecasting to the suit of tasks supported by `ClimateLearn`. We implemented two variants of the ResNet model for this task: ResNet-parametric which directly outputs the mean and standard deviation of a Gaussian distribution as the prediction of the future, and ResNet-ensemble where we train 10 different instances of the same ResNet architecture using different seeds and averaging the predictions of these instances to get the final prediction. We evaluate these two baselines using RMSE, ACC, and the probabilistic metrics CRPS and Spread-skill ratio introduced in [21]. Table 4 compares the two baselines in this task.

# 6 Conclusion

We presented `ClimateLearn`, a user-friendly and open-source PyTorch library for data-driven weather and climate modeling. Given the pressing nature of climate change, we believe our contribution is timely and of potential use to both the ML and climate science communities. Our objective is to provide a standardized benchmarking platform for evaluating ML innovations in climate science, which currently suffer from challenges in standardization, accessibility, and reproducibility. `ClimateLearn` provides users access to all essential components of end-to-end ML pipeline, including data pre-processing utilities, ML model implementations, and rigorous evaluations via metrics and visualizations. We use the flexible and modular design of `ClimateLearn` to design and perform diverse experiments comparing deep learning methods with relevant baselines on our supported tasks.

**Limitations and Future Work** In this work, we highlighted key features of the `ClimateLearn` library, encompassing datasets, tasks, models, and evaluations. However, we acknowledge that there are numerous avenues to enhance the comprehensiveness of our library in each of these dimensions. One such avenue involves integrating regional datasets and expanding the catalog of available data sources. On the modeling side, we plan to develop efficient implementations for training ensembles in service of critical uncertainty quantification efforts. In future iterations of our library, we will also integrate a hub of large-scale pretrained neural networks specifically designed for weather and climate applications [4, 38, 52, 54]. Once integrated, these pretrained models will be further customizable through fine-tuning, enabling straightforward adaptation to downstream tasks. Furthermore, we plan to incorporate support for physics-informed neural networks and other hybrid baselines that amalgamate physical models with machine learning methods, which will allow users to leverage the strengths of both paradigms. Ultimately, our overarching objective is to establish `ClimateLearn` as a trustworthy AI development tool for weather and climate applications.

# Acknowledgments and Disclosure of Funding

We thank World Climate Research Programme (WCRP) for the CMIP6 data collection, and European Center for Medium-Range Weather Forecasting (ECMWF) for the ERA5 dataset. We thank Shashank Goel, Jingchen Tang, Seongbin Park, Siddharth Nandy, and Sri Keerthi Bolli for their contributions to `ClimateLearn`. Aditya Grover was supported in part by a research gift from Google. Hritik Bansal was supported in part by AFOSR MURI grant FA9550-22-1-0380.

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
