## A    Licenses and Terms of Use

ClimateLearn is a software package that can be installed from the Python Package Index as follows.

```
pip install climate-learn
```

The source code is available online under the MIT License at `https://github.com/aditya-grover/climate-learn`, and the accompanying documentation website is at `https://climatelearn.readthedocs.io/`. The Extreme-ERA5 dataset does not exist as a distinct entity, but can be produced by running code provided in our library. The Machine Intelligence Group at UCLA is the maintainer of ClimateLearn.

The sources for datasets provided by ClimateLearn are WeatherBench, ClimateBench, the Earth System Grid Federation (ESGF), the Copernicus Climate Data Store (CDS), and PRISM. The WeatherBench dataset (`https://mediatum.ub.tum.de/1524895`), ClimateBench dataset (`https://zenodo.org/record/7064308`), and MPI-ESM1.2-HR outputs from ESGF (`https://pcmdi.llnl.gov/CMIP6/TermsOfUse/TermsOfUse6-2.html`) are available under the CC BY 4.0 license. Neither Copernicus (`https://cds.climate.copernicus.eu/api/v2/terms/static/licence-to-use-copernicus-products.pdf`) nor PRISM (`https://prism.oregonstate.edu/terms/`) use a Creative Commons License. Instead, they each set forth their own terms of use, both of which permit reproduction and distribution for non-commercial purposes.

## B    Experiment details

### B.1    Network architectures

#### B.1.1    ResNet

Our ResNet architecture is similar to that of WeatherBench [61, 60], in which each residual block consists of two identical convolutional modules: 2D convolution $\rightarrow$ LeakyReLU with $\alpha = 0.3 \rightarrow$ Batch Normalization $\rightarrow$ Dropout.

Table 4: Default hyperparameters of ResNet

| Hyperparameter | Meaning | Value |
| --- | --- | --- |
| Padding size | Padding size of each convolution layer | 1 |
| Kernel size | Kernel size of each convolution layer | 3 |
| Stride | Stride of each convolution layer | 1 |
| Hidden dimension | Number of output channels of each residual block | 128 |
| Residual blocks | Number of residual blocks | 28 |
| Dropout | Dropout rate | 0.1 |

Table 4 shows the hyperparameters for ResNet in all of our experiments. We use a convolutional layer with a kernel size of 7 at the beginning of the network. All paddings are periodic in the longitude direction and zeros in the latitude direction.

#### B.1.2    UNet

We borrow our UNet implementation from PDEArena [21]. Table 5 shows the hyperparameters for UNet in all of our experiments. Similar to ResNet, we use a convolutional layer with a kernel size of 7 at the beginning of the network, and all paddings are periodic in the longitude direction and zeros in the latitude direction.

Table 5: Default hyperparameters of UNet

| Hyperparameter | Meaning | Value |
|---|---|---|
| Padding size | Padding size of each convolution layer | 1 |
| Kernel size | Kernel size of each convolution layer | 3 |
| Stride | Stride of each convolution layer | 1 |
| Hidden dimension | Base number of output channels | 64 |
| Channel multiplications | Determine the number of output channels for Down and Up blocks | $[1, 2, 2]$ |
| Blocks | Number of blocks | 2 |
| Use attention | If use attention in Down and Up blocks | False |
| Dropout | Dropout rate | 0.1 |

### B.1.3 ViT

We use the standard Vision Transformer architecture [14] with minor modifications. We remove the class token and add a 1-hidden MLP prediction head which is applied to the tokens after the last attention layer to predict the outputs. Tabel 6 shows the hyperparameters for ViT in all of our experiments.

Table 6: Default hyperparameters of ViT

| Hyperparameter | Meaning | Value |
|---|---|---|
| $p$ | Patch size | 2 |
| $D$ | Embedding dimension | 128 |
| Depth | Number of ViT blocks | 8 |
| # heads | Number of attention heads | 4 |
| MLP ratio | Determine the hidden dimension of the MLP layer in a ViT block | 4 |
| Prediction depth | Number of layers of the prediction head | 2 |
| Hidden dimension | Hidden dimension of the prediction head | 128 |
| Drop path | For stochastic depth [30] | 0.1 |
| Dropout | Dropout rate | 0.1 |

### B.2 Datasets

### B.2.1 ERA5

We refer to `https://confluence.ecmwf.int/display/CKB/ERA5%3A+data+documentation` for more details of the raw ERA5 data. We use the preprocessed version of ERA5 at $5.625°$ from WeatherBench [61] for our experiments. Table 7 summarizes the variables we use for our experiments.

### B.2.2 Extreme-ERA5

**Calculating thresholds** We use the surface temperature (T2m) data corresponding to the years $1979 - 2015$ from ERA5 at a resolution of $5.625°$ to calculate the thresholds. The thresholds are localized i.e. they are calculated for every pixel on the grid. For a given timestamp and pixel, we first calculate a 7 day mean till that timestamp. Now, to account for neighboring regions/pixels, we set the localized mean as 0.44 * current pixel's mean + 0.11 * sum of means of pixels sharing an edge + 0.027 * sum of means of pixels sharing a vertex but not an edge. Note, that there is no need of padding while accounting for neighboring pixels, since earth is a globe, the neighbors of leftmost pixels include the rightmost pixels and vice-versa. Finally, the 5th and 95th percentile values of this new mean data corresponding to every pixel is set as threshold.

Table 7: ERA5 variables used in our experiments. *Constant* represents constant variables, *Single* represents surface variables, and *Atmospheric* represents atmospheric properties at the chosen altitudes.

| Type | Variable name | Abbrev. | Levels |
|---|---|---|---|
| Static | Land-sea mask | LSM | |
| Static | Orography | | |
| Static | Latitude | | |
| Single | Toa incident solar radiation | Tisr | |
| Single | 2 metre temperature | T2m | |
| Single | 10 metre U wind component | U10 | |
| Single | 10 metre V wind component | V10 | |
| Atmospheric | Geopotential | Z | 50, 250, 500, 600, 700, 850, 925 |
| Atmospheric | U wind component | U | 50, 250, 500, 600, 700, 850, 925 |
| Atmospheric | V wind component | V | 50, 250, 500, 600, 700, 850, 925 |
| Atmospheric | Temperature | T | 50, 250, 500, 600, 700, 850, 925 |
| Atmospheric | Specific humidity | Q | 50, 250, 500, 600, 700, 850, 925 |
| Atmospheric | Relative humidity | R | 50, 250, 500, 600, 700, 850, 925 |

**Building masks** As the purpose of Extreme-ERA5 is evaluation of forecasting models under extreme weather conditions, we build it for test years i.e. $2017 - 2018$ only. We first create a 2-D mask of size, latitude x longitude, filled with zeros for every available timestamp in the test years. Similar to the calculating thresholds, we compute the mean of each pixel at every timestamp for T2m's test data. We then, set the value for a given pixel in the mask as 1, if the mean value is outside the bounds set by the thresholds. Finally, during evaluation time, we use these masks to select subset of data.

### B.2.3 CMIP6

**MPI-ESM1.2-HR** We use MPI-ESM1.2-HR, a dataset in the CMIP6 data repository for our experiments in Section 4.1.3. Table 8 summarizes the variables we use for our experiments.

Table 8: MPI-ESM1.2-HR variables used in our experiments. *Single* represents surface variables and *Atmospheric* represents atmospheric properties at the chosen altitudes.

| Type | Variable name | Abbrev. | Levels |
|---|---|---|---|
| Single | 2 metre temperature | T2m | |
| Single | 10 metre U wind component | U10 | |
| Single | 10 metre V wind component | V10 | |
| Atmospheric | Geopotential | Z | 50, 250, 500, 600, 700, 850, 925 |
| Atmospheric | U wind component | U | 50, 250, 500, 600, 700, 850, 925 |
| Atmospheric | V wind component | V | 50, 250, 500, 600, 700, 850, 925 |
| Atmospheric | Temperature | T | 50, 250, 500, 600, 700, 850, 925 |
| Atmospheric | Specific humidity | Q | 50, 250, 500, 600, 700, 850, 925 |

**ClimateBench** We adopt data from ClimateBench [82] for our climate projection experiment. ClimateBench contains simulated data from experimental runs by the Norwegian Earth System Model [69], a member of CMIP6, on different emission scenarios. Specifically, ClimateBench includes 7 esmission scenarios: historical, ssp126, ssp370, ssp585, hist-aer, hist-GHG, and ssp245. We refer to the original ClimateBench paper for the exact temporal coverage and more details of these scenarios.

### B.3 Training details

#### B.3.1 Continuous training

Continuous models additionally condition on lead times to make predictions. To do this, we add the lead time value in hours divided by 100 to the input channels to make the model aware of the lead time it is forecasting at. During training, we randomize the lead time from 6 hours to 5 days $\Delta t \sim \mathcal{U}[6, 120]$, and during evaluation, we fix the lead time to a certain value to evaluate the model's performance at a certain lead time. This setting was commonly used in previous works [50, 61].

#### B.3.2 Software and hardware stack

We use PyTorch [51], `numpy` [24] and `xarray` [29] to manage our data and model training. We also use `timm` [86] for our ViT implementation. All training is done on 10 AMD EPYC 7313 CPU cores and one NVIDIA RTX A5000 GPU. We leverage `fp16` floating point precision in our experiments.

### B.4 Metrics

We use the following definitions in our metric formulations

- $N$ is the number of data points
- $H$ is the number of latitude coordinates.
- $W$ is the number of longitude coordinates.
- $X$ and $\tilde{X}$ are the ground-truth and prediction, respectively.

The latitude weighting function is given by

$$L(i) = \frac{\cos(H_i)}{\frac{1}{H} \sum_{i=1}^{H} \cos(H_i)} \tag{1}$$

#### B.4.1 Deterministic weather forecasting metrics

**Root mean square error (RMSE)**

$$\text{RMSE} = \frac{1}{N} \sum_{k=1}^{N} \sqrt{\frac{1}{H \times W} \sum_{i=1}^{H} \sum_{j=1}^{W} L(i)(\tilde{X}_{k,i,j} - X_{k,i,j})^2}. \tag{2}$$

**Anomaly correlation coefficient (ACC)** is the spatial correlation between prediction anomalies $\tilde{X}'$ relative to climatology and ground truth anomalies $X'$ relative to climatology:

$$\text{ACC} = \frac{\sum_{k,i,j} L(i)\tilde{X}'_{k,i,j} X'_{k,i,j}}{\sqrt{\sum_{k,i,j} L(i)\tilde{X}'^2_{k,i,j} \sum_{k,i,j} L(i)X'^2_{k,i,j}}}, \tag{3}$$

$$\tilde{X}' = \tilde{X}' - C, X' = X' - C, \tag{4}$$

in which climatology $C$ is the temporal mean of the ground truth data over the entire test set $C = \frac{1}{N} \sum_{k} X$.

#### B.4.2 Probabilistic weather forecasting metrics

**Spread-skill ratio (Spread by RMSE)** measures a probabilistic forecast's reliability. Let $N$ be the number of forecasts produced either by ensembling or drawing samples from a parametric prediction. Spread is given by

$$\text{Spread} = \frac{1}{N} \sum_{k}^{N} \sqrt{\frac{1}{H \times W} \sum_{i=1}^{H} \sum_{j=1}^{W} L(i)\text{var}(\tilde{X}_{i,j})} \tag{5}$$

**Continuous ranked probability score** measures a probabilistic forecast's calibration and sharpness. Let $F$ denote the CDF of the forecast distribution. For a Gaussian distribution parameterized by mean $\mu$ and standard deviation $\sigma$, the closed-form, differentiable solution is

$$\text{CRPS}(F_{\mu,\sigma}, X) = \sigma \left\{ \frac{X-\mu}{\sigma} \left[ 2\Phi\left(\frac{X-\mu}{\sigma}\right) - 1 \right] + 2\phi\left(\frac{X-\mu}{\sigma}\right) - \frac{1}{\sqrt{\pi}} \right\} \tag{6}$$

where $\Phi$ and $\phi$ are the CDF and PDF of the standard normal distribution, respectively.

### B.4.3 Climate downscaling metrics

**Root mean square error (RMSE)** This is the same as Equation (2).

**Mean bias** measures the difference between the spatial mean of the prediction and the spatial mean of the ground truth. A positive mean bias shows an overestimation, while a negative mean bias shows an underestimation of the mean value.

$$\text{Mean bias} = \frac{1}{N \times H \times W} \sum_{k=1}^{N} \sum_{i=1}^{H} \sum_{j=1}^{W} \tilde{X} - \frac{1}{N \times H \times W} \sum_{k=1}^{N} \sum_{i=1}^{H} \sum_{j=1}^{W} X \tag{7}$$

**Pearson coefficient** measures the correlation between the prediction and the ground truth. We first flatten the prediction and ground truth, and compute the metric as follows:

$$\rho_{\tilde{X},X} = \frac{\text{cov}(\tilde{X}, X)}{\sigma_{\tilde{X}} \sigma_X} \tag{8}$$

**Masking for PRISM** Since PRISM does not record data over the oceans, we mask out those values for evaluation. Concretely, we set `NaN` values in the ground truth data to $0$. Then, we multiply the model's predictions by a binary mask that is $0$ wherever the ground truth data is originally `NaN` and is $1$ everywhere else.

### B.4.4 Climate projection metrics

**Normalized spatial root mean square error (NRMSE$_s$)** measures the spatial discrepancy between the temporal mean of the prediction and the temporal mean of the ground truth:

$$\text{NRMSE}_s = \sqrt{\left\langle \left( \frac{1}{N} \sum_{k=1}^{N} \tilde{X} - \frac{1}{N} \sum_{k=1}^{N} X \right)^2 \right\rangle} \bigg/ \frac{1}{N} \sum_{k=1}^{N} \langle X \rangle, \tag{9}$$

in which $\langle A \rangle$ is the global mean of $A$:

$$\langle A \rangle = \frac{1}{H \times W} \sum_{i=1}^{H} \sum_{j=1}^{W} L(i) A_{i,j} \tag{10}$$

**Normalized global root mean square error (NRMSE$_g$)** measures the discrepancy between the global mean of the prediction and the global mean of the ground truth:

$$\text{NRMSE}_g = \sqrt{\frac{1}{N} \sum_{k=1}^{N} \left( \langle \tilde{X} \rangle - \langle X \rangle \right)^2} \bigg/ \frac{1}{N} \sum_{k=1}^{N} \langle X \rangle. \tag{11}$$

**Total normalized root mean square error (Total)** is the weighted sum of NRMSE$_s$ and NRMSE$_g$:

$$\text{Total} = \text{NRMSE}_s + \alpha \cdot \text{NRMSE}_g, \tag{12}$$

where $\alpha$ is chosen to be $5$ as suggested by Watson-Parris et al. [82].

## C Additional experiments

### C.1 Climate projection

**Task** We consider the task of predicting the annual mean distributions of 4 target variables in ClimateBench [82]: surface temperature, diurnal temperature range, precipitation, and the 90th percentile of precipitation.

**Baselines** We compare ResNet, UNet, and ViT, three deep learning models supported by `ClimateLearn` with CNN-LSTM, the deep learning baseline in ClimateBench. The network architectures of the three models are identical to Appendix B.1.

**Data** We regrid the original ClimateBench data to $5.625°$ for easy training and evaluation. The input variables include 4 forcing factors: carbon dioxide ($CO_2$), sulfur dioxide ($SO_2$), black carbon (BC), and methane ($CH_4$). Similar to the deep learning baseline in ClimateBench, we stack 10 consecutive years to predict the target variables of the current year. We standardize the input channels to have 0 mean and 1 standard deviation, but do not standardize the output variables. Training and validation data includes the historical data, ssp126, ssp370, ssp585, and the historical data with aerosol (hist-aer) and greenhouse gas (hist-GHG) forcings, and test data includes ssp245. We split train/validation data with a ratio of $0.9/0.1$.

**Training and evaluation** We train one network for each target variable. We use the same optimizer and scheduler as in Section 4.1. We train for 50 epochs with 16 batch size, and use early stopping with a patience of 5 epochs. We use mean-squared error as the loss function and evaluation metric. We report normalized spatial root mean square error ($NRMSE_s$), normalized global root mean square error ($NRMSE_g$), and Total = $NRMSE_s + 5×NRMSE_g$ as test metrics.

**Results** Table 9 shows the performance of different baselines on ClimateBench. CNN-LSTM and UNet are the best-performing methods, with each achieving the best performance in $5/12$ metrics, followed by ResNet which performs best on $2/12$ metrics. ViT achieves a reasonable performance but underperforms the CNN-based methods.

Table 9: Performance of different deep learning baselines on ClimateBench. CNN-LSTM result is taken from ClimateBench.

| | Surface temperature | | | Diurnal temperature range | | | Precipitation | | | 90th percentile precipitation | | |
|---|---|---|---|---|---|---|---|---|---|---|---|---|
| | $NRMSE_s$ | $NRMSE_g$ | Total | $NRMSE_s$ | $NRMSE_g$ | Total | $NRMSE_s$ | $NRMSE_g$ | Total | $NRMSE_s$ | $NRMSE_g$ | Total |
| CNN-LSTM | 0.107 | 0.044 | **0.327** | 9.917 | 1.372 | 16.778 | **2.128** | 0.209 | **3.175** | **2.610** | 0.346 | **4.339** |
| ResNet | 0.182 | **0.042** | 0.395 | 9.128 | **0.737** | 12.810 | 2.930 | 0.180 | 3.828 | 3.413 | 0.286 | 4.845 |
| UNet | **0.097** | 0.046 | 0.328 | **6.300** | 0.946 | **11.030** | 2.483 | **0.141** | 3.187 | 3.122 | **0.282** | 4.532 |
| ViT | 0.191 | 0.092 | 0.650 | 7.725 | 0.746 | 11.460 | 2.909 | 0.327 | 4.545 | 3.615 | 0.418 | 5.704 |

### C.2 Extreme weather prediction

Table 10 shows the performance of different models across various different lead times on the default test split and Extreme-ERA5. As discussed in Section 4.1.2, the performance of all models except Climatology is better on the extreme split than on the default split.

Table 10: Latitude-weighted RMSE on the normal and extreme test splits of ERA5 for different lead times.

| T2M | 6 Hours | 1 Day | 3 Days | 5 Days | 10 Days |
|---|---|---|---|---|---|
| Climatology | 5.87 / 6.51 | 5.87 / 6.53 | 5.87 / 6.58 | 5.88 / 6.65 | 5.89 / 6.76 |
| Persistence | 2.76 / 2.99 | 2.13 / 1.78 | 2.99 / 2.42 | 3.26 / 2.61 | 3.59 / 2.89 |
| ResNet | 0.72 / **0.72** | 0.94 / **0.91** | 1.50 / **1.33** | 2.20 / **1.86** | 2.78 / 2.39 |
| U-Net | 0.76 / 0.77 | 1.04 / 0.99 | 1.65 / 1.43 | 2.26 / 1.88 | 2.76 / 2.44 |
| ViT | 0.78 / 0.80 | 1.09 / 1.05 | 1.71 / 1.55 | 2.38 / 2.04 | 2.78 / **2.30** |

Table 11: Performance of different models trained on one dataset (columns) and evaluated on another (rows). Training data for CMIP6 is available from the years $1850 - 2010$, at a 6 hour frequency. Training data for ERA5 is available from years $1979 - 2010$, at an one hour frequency.

| | | | 3 Days | | | | 5 Days | | | |
| | | | ERA5 | | CMIP6 | | ERA5 | | CMIP6 | |
| | | | ACC | RMSE | ACC | RMSE | ACC | RMSE | ACC | RMSE |
|---|---|---|---|---|---|---|---|---|---|---|
| ERA5 | ResNet | Z500 | 0.95 | 315.07 | **0.96** | **302.08** | 0.77 | 646.57 | **0.86** | **531.47** |
| | | T850 | **0.93** | **1.84** | 0.91 | 2.08 | 0.80 | 3.00 | **0.83** | **2.77** |
| | | T2m | **0.95** | **1.56** | 0.94 | 1.85 | 0.89 | 2.35 | **0.90** | **2.29** |
| | U-Net | Z500 | 0.92 | 388.17 | **0.94** | **337.34** | 0.74 | 686.90 | **0.82** | **590.80** |
| | | T850 | **0.91** | **2.09** | 0.90 | 2.17 | 0.78 | 3.10 | **0.81** | **2.93** |
| | | T2m | **0.95** | **1.72** | 0.93 | 1.89 | 0.89 | 2.38 | **0.89** | **2.37** |
| | ViT | Z500 | 0.93 | 380.22 | **0.93** | **373.57** | 0.68 | 749.82 | **0.82** | **592.36** |
| | | T850 | **0.91** | **2.08** | 0.89 | 2.31 | 0.75 | 3.27 | **0.80** | **2.97** |
| | | T2m | **0.94** | **1.73** | 0.92 | 2.10 | 0.88 | 2.54 | **0.88** | **2.52** |
| CMIP6 | ResNet | Z500 | 0.95 | 35.84 | **0.98** | **24.51** | 0.77 | 71.50 | **0.89** | **50.58** |
| | | T850 | 0.92 | 2.09 | **0.96** | **1.43** | 0.79 | 3.19 | **0.89** | **2.33** |
| | | T2m | 0.94 | 1.88 | **0.97** | **1.32** | 0.88 | 2.54 | **0.94** | **1.87** |
| | U-Net | Z500 | 0.92 | 43.36 | **0.96** | **30.61** | 0.75 | 74.68 | **0.85** | **58.67** |
| | | T850 | 0.90 | 2.30 | **0.95** | **1.67** | 0.78 | 3.29 | **0.87** | **2.57** |
| | | T2m | 0.93 | 2.00 | **0.96** | **1.46** | 0.88 | 2.57 | **0.93** | **1.99** |
| | ViT | Z500 | 0.93 | 42.19 | **0.95** | **34.83** | 0.68 | 83.68 | **0.85** | **58.86** |
| | | T850 | 0.90 | 2.25 | **0.94** | **1.83** | 0.75 | 3.48 | **0.86** | **2.60** |
| | | T2m | 0.92 | 2.15 | **0.95** | **1.59** | 0.85 | 2.88 | **0.92** | **2.03** |

Table 12: Performance of different models trained on one dataset (columns) and evaluated on another (rows). The training years and data availability frequency is same for both the datasets.

| | | | 3 Days | | | | 5 Days | | | |
| | | | ERA5 | | CMIP6 | | ERA5 | | CMIP6 | |
| | | | ACC | RMSE | ACC | RMSE | ACC | RMSE | ACC | RMSE |
|---|---|---|---|---|---|---|---|---|---|---|
| ERA5 | ResNet | Z500 | **0.95** | **322.86** | 0.94 | 345.00 | **0.79** | **624.20** | 0.78 | 646.48 |
| | | T850 | **0.93** | **1.90** | 0.90 | 2.21 | **0.81** | **2.91** | 0.79 | 3.11 |
| | | T2m | **0.95** | **1.62** | 0.93 | 1.94 | **0.90** | **2.33** | 0.88 | 2.55 |
| | U-Net | Z500 | **0.92** | **401.08** | 0.91 | 422.77 | **0.74** | **685.75** | 0.73 | 712.62 |
| | | T850 | **0.90** | **2.17** | 0.88 | 2.42 | **0.78** | **3.10** | 0.76 | 3.29 |
| | | T2m | **0.94** | **1.81** | 0.91 | 2.19 | **0.89** | **2.44** | 0.86 | 2.73 |
| | ViT | Z500 | **0.91** | **426.70** | 0.90 | 444.14 | **0.72** | **698.08** | 0.72 | 720.15 |
| | | T850 | **0.89** | **2.27** | 0.87 | 2.51 | **0.78** | **3.12** | 0.76 | 3.31 |
| | | T2m | **0.94** | **1.88** | 0.91 | 2.19 | **0.89** | **2.43** | 0.87 | 2.69 |
| CMIP6 | ResNet | Z500 | 0.95 | 36.47 | **0.96** | **31.29** | 0.79 | 69.62 | **0.81** | **65.04** |
| | | T850 | 0.91 | 2.11 | **0.94** | **1.70** | 0.81 | 3.09 | **0.84** | **2.82** |
| | | T2m | 0.93 | 1.91 | **0.96** | **1.53** | 0.88 | 2.51 | **0.91** | **2.24** |
| | U-Net | Z500 | 0.92 | 44.95 | **0.93** | **40.98** | 0.74 | 75.45 | **0.76** | **72.67** |
| | | T850 | 0.89 | 2.37 | **0.92** | **2.05** | 0.78 | 3.27 | **0.81** | **3.04** |
| | | T2m | 0.91 | 2.23 | **0.94** | **1.74** | 0.86 | 2.73 | **0.90** | **2.31** |
| | ViT | Z500 | 0.91 | 46.92 | **0.92** | **43.91** | 0.73 | 76.80 | **0.75** | **74.22** |
| | | T850 | 0.89 | 2.40 | **0.91** | **2.15** | 0.77 | 3.29 | **0.80** | **3.06** |
| | | T2m | 0.91 | 2.15 | **0.94** | **1.82** | 0.87 | 2.68 | **0.90** | **2.32** |

## C.3 Dataset robustness

Table 11 shows the comparison of the performance for different models when trained on ERA5 and evaluated on CMIP6 and vice versa at 3 and 5 days of lead time. For the CMIP6 evaluation purposes, the models trained on ERA5 were slightly worse than the moodels trained on CMIP6. Surprisingly, for evaluating on ERA5, models trained on CMIP6 were comparable, if not slightly better to the ones trained on ERA5. These results are in line with results of [50], thus highlighting the dataset usefulness of CMIP6 over ERA5. Note that the data's raw size is roughly similar for both the datasets as despite the ERA5's temporal training range being 1979-2010 in this setup, it's data availability frequency is 1 hour compared to 6 hour in CMIP6.

To find out whether this superiority of CMIP6 over ERA5 is just a result of differences in temporal range, we conducted the similar study but with same dataset temporal characteristics (i.e. setting training years as 1979-2010 and subsampling the data at 6 hours). This time the results just for ResNet at 3 day lead time is shown in Table 2 and for all models at different lead times, is shown in Table 12. These results show that the performance is slightly worse for both the cases now. Thus showing that the performance improvement of training over CMIP6 than ERA5 is likely just the bigger temporal range.

## D   Visualizations

`ClimateLearn` provides visualization functionality to help with an intuitive understanding of model performance. Below is an example figure generated by `ClimateLearn` for visualizing the quality of a model's forecast. Each row represents a distinct time in the test set. The leftmost column shows weather conditions at the time the model is making a prediction from. The next column shows the ground truth conditions at the forecast horizon. The next column shows the model's predictions. The last column shows the model's bias, and its per-pixel forecast error.

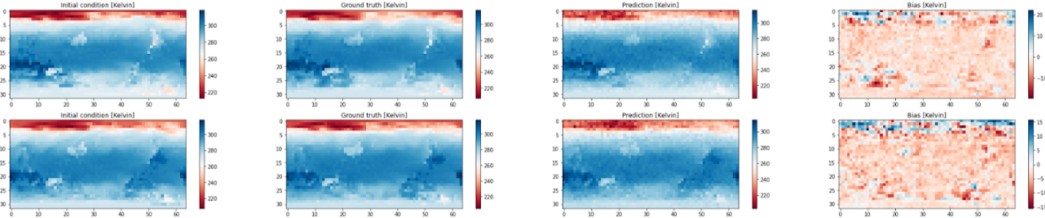

Figure 4: Example visualization of deterministic forecasting.

Additionally, `ClimateLearn` can generate the rank histogram for probabilistic forecasts. A rank histogram that resembles a uniform distribution means that the ground truth value is indistinguishable from any member of the forecast ensemble. A rank histogram that is skew right occurs when the ground truth is consistently lower than the ensemble prediction. A rank histogram that appears U-shaped is indicative of both low biases and high biases. An example figure generated by `ClimateLearn` for visualizing the rank histogram is shown below.

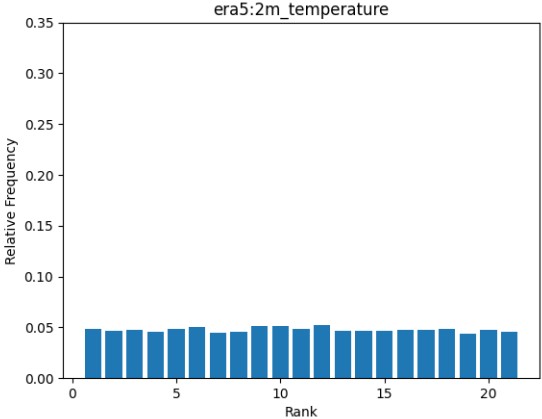

Figure 5: Example visualization of the rank histogram for probabilistic forecasting.

We show an example of how to generate another visualization called "mean-bias" in the next section.

## E  Code snippets

ClimateLearn can be used to download heterogeneous climate data from a variety of sources in a single function call. Here, we provide an example for downloading ERA5 2-meter temperature data at $5.625°$ resolution from WeatherBench.

```python
from climate_learn.data import download
download(
    root="./weatherbench-data",
    source="weatherbench",
    dataset="era5",
    resolution="5.625",
    variable="2m_temperature"
)
```

Further, ClimateLearn can process downloaded data into a form that is loadable into PyTorch. In fewer than **30 lines**, the following code loads raw ERA5 data; normalizes it; splits it into train, validation, testing sets; and prepares batches for the forecasting task.

```python
# For flexibility with loading datasets and implementing new ones
# in the future, ClimateLearn's data processing pipeline is made
# up of three parts: the climate dataset (e.g., ERA5), the task
# (e.g., forecasting), and the PyTorch dataset (e.g., Map). These
# are all combined into a Pytorch Lightning DataModule.
from climate_learn.data.climate_dataset.args import ERA5Args
from climate_learn.data.task.args import ForecastingArgs
from climate_learn.data.dataset import MapDatasetArgs
from climate_learn.data import DataModule

# Next, we define the arguments: the location of the data, the
# variables we will use as input/target, and the data splits
root        = "./weatherbench-data"
variables   = ["2m_temperature"]
train_years = range(1979, 2016)
val_years   = range(2016, 2017)
test_years  = range(2017, 2019)

# Next, we construct the arguments for the three parts of the data
# processing pipeline to build the training dataset.
climate_dataset_args = ERA5Args(root, variables, train_years)
task_args = ForecastingArgs(
    [f"era5:{var}" for var in variables], # format - dataset:var
    [f"era5:{var}" for var in variables], # format - dataset:var
    pred_range=72,                        # hours ahead to predict
    history=3,                            # past time steps
    subsample=6                           # hours per time step
)
train_data_args = MapDatasetArgs(climate_dataset_args, task_args)

# The validation and test datasets can be constructed easily by
# copying arguments from the train dataset which are the same and
# modifying only what is needed.
```

```
34    val_data_args = train_data_args.create_copy({
35        "climate_dataset_args": {"years": val_years}
36    })
37    test_data_args = val_data_args.create_copy({
38        "climate_dataset_args": {"years": test_years}
39    })
40
41    # Finally, we can unify all parts of the data pipeline to get a
42    # single PyTorch Lightning data module.
43    dm = DataModule(train_data_args, val_data_args, test_data_args)
```

With the loaded data, `ClimateLearn` can be used to build, train, and evaluate a model in fewer than **20 lines** of code.

```
1    import climate_learn as cl
2    from climate_learn.training import Trainer
3
4    model_kwargs = {
5        "in_channels": 1,   # predicting 2m_temperature
6        "history": 3,       # matching 'ForecastingArgs'
7        "n_blocks": 4       # number of residual blocks to use
8    }
9    optim_kwargs = {}       # use the default settings
10   mm = cl.load_forecasting_module(
11       data_module=dm,
12       model="resnet",
13       model_kwargs=model_kwargs,
14       optim_kwargs=optim_kwargs
15   )
16
17   trainer = Trainer()
18   trainer.fit(mm, dm)
19   trainer.test(mm, dm)
20
```

`ClimateLearn` can also be used to load pre-defined models (e.g., persistence, Rasp and Thuerey [60]) as follows.

```
1    persistence = cl.load_forecasting_module(
2        data_module=dm,
3        preset="persistence"
4    )
5    rasp_theurey_2020 = cl.load_forecasting_module(
6        data_module=dm,
7        preset="rasp-theurey-2020"
8    )
```

`ClimateLearn` can use the trained forecasting models to produce visualizations in a single line of code. For example, one visualization of interest is the mean bias, which shows the expected error of the model's forecast, per pixel, over the evaluation period.

```
1  from climate_learn.utils.visualize import visualize_mean_bias
2  visualize_mean_bias(persistence, dm)
```

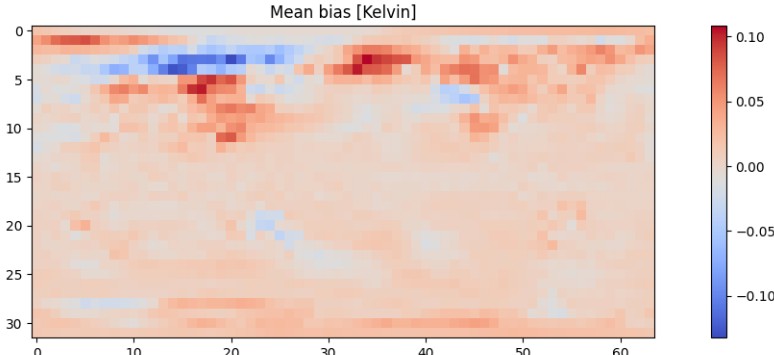

Figure 6: Visualization of the mean bias of temperature

This graphic shows that, on average, persistence has little bias below the equator. Over the northern part of North America, persistence achieves negative mean bias, which means it generally under-predicts 2-meter temperature in that region. Meanwhile, in the northern part of Europe, persistence achieves positives mean bias, indicating overprediction.