# OpenReview forum: "ClimateLearn: Benchmarking Machine Learning for Weather and Climate Modeling"
_NeurIPS.cc/2023/Track/Datasets_and_Benchmarks — NeurIPS 2023 Datasets and Benchmarks Poster_

### Official Review · Reviewer_T3D2 · 2023-07-21
**Review for ClimateLearn, a library of models and datasets for weather and climate emulation by deep learning models**

**Rating:** 6
**Confidence:** 4
**Correctness:** Yes
**Clarity:** Paper is well-written.

**Strengths:**

The main strength is that the authors collect and curate in the same place different DL models as well as datasets (and associated tasks) that acts as a benchmark of ML researchers to push the frontier -- instead of different benchmarks for different tasks (weather forecasting, extreme events, climate emulation, downscaling etc), this library aims to bring everything under one roof with ease of access.

**Additional Feedback:**

Additional comments:
1. My main issue with the paper is that a large part of it is just reproducing other benchmarks and placing them all in a new repository without further modifications. While I appreciate the engineering effort and recognize the usefulness of a single repo containing all, I am unsure about the paper. For example, ERA5 forecasting: today many papers/groups have come up with models at higher resolution (25km/0.25deg, open-sourced through several venues) with large variable sets demonstrating capabilities of ML (some even beating IFS due to training on re-analysis) such as GraphCast, PanguWeather, FengWu, etc. However, this paper sticks to the resolution of the old weatherbench paper/repo (which is quite coarse) and reproduces all their results. It is hard to justify benchmarking ML models at this coarse scale today (especially for extreme events which need much higher resolutions) and continue comparing these to an NWP model (that was probably run at <=10km scale and is further improving). While weatherbench et al. was seminal when it came out, it does seem insufficient to not improve upon it when many research teams already benchmark at higher resolutions.
2. A related point to above is that today climate models are moving even further in resolution to better resolve certain processes and benchmarking ML models at 22x (and lower) resolutions might not be the most useful direction.
3. The data robustness to distributional shift is interesting and I have seen it in other papers as well w.r.t ERA5 and CMIP6 (including weatherbench). It would be useful to show another dataset from a climate model simulation. In Tab 2, what is the lead time?
4. Question: The authors mention PINNs which is an interesting new area of research in approximating the solution of PDEs with NNs. While I am not an NWP researcher, it is my understanding that weather/climate models contain many PDEs and parameterizations to close the system. Is the goal to include all these equations as a loss penalty as in PINNs?
5. How do the authors compare their work with modulus-launch from NVIDIA which aims to do the same thing (but also across other disciplines): training recipes, models and datasets for AI for science, and includes weather forecasting.
6. In the downscaling experiments, what is the ground truth for the high resolution variables? Do they just come from interpolated variables or is it from newer physics (such as pde solvers run on a higher resolution grid)?

**Documentation:**

Yes.

**Ethics:**

None.

**Limitations:**

Authors talk about better training recipes, pretraining (in hopes of possibly yielding foundation models in the future) recipes, PINNs etc as current limitations. However, I feel resolution and variable set limitations are far more important than these at the moment.

**Opportunities For Improvement:**

- Datasets are all downsampled to coarse resolutions (even the fields in the appendix for visualizations look very blocky) and at such resolutions have limited utility to scientists.
- Related to above, variable sets are also small (for instance, ERA5 has a large variable set of 100s of state variables and it has been shown in recent papers that ML models do much better as more variables are added to the list, as expected from data-driven models).


**Relation To Prior Work:**

Yes.

**Summary And Contributions:**

The authors present an open-source PyTorch library of models and associated datasets for evaluating deep learning models on different weather and climate related tasks. The main contributions are an end-to-end pipeline of models and datasets that will democratize ML research in this field, with an inclusion of important scientific tasks as benchmarks to build better models.

---

> ### Author Response · Authors · 2023-08-22
>
> We thank the reviewer for the constructive feedback and for the recognition of the contributions of ClimateLearn. We have updated the paper to show more experiments and new features of the package, as well as to clarify certain points in the paper. We answer each of the reviewer's concerns below.
>
> > Datasets are all downsampled to coarse resolutions and at such resolutions have limited utility to scientists.
>
> We used coarse-resolution data in the experiments because of our limited storage and compute. The 0.25deg data with the same number of variables we consider in the experiments is about 148 Terabytes, so it’s not trivial to download/store data and run models at this resolution in an academic setting. However, we note that ClimateLearn also supports higher-resolution data if one has the compute to do so. For example, one can download the 2m_temperature variable of the original 0.25deg ERA5 from Corpenicus using ClimateLearn with the following line of code:
> ```
> climate_learn.data.download(
> 	source=’copernicus’,
> 	dataset=’era5’,
> 	root=’/home/data/era5/’,
>     variable=’2m_temperature’,
> 	year=2020
> )
> ```
> > Variable sets are also small (for instance, ERA5 has a large variable set of 100s of state variables and it has been shown in recent papers that ML models do much better as more variables are added to the list, as expected from data-driven models)
>
> We only included a small set of variables in the experiments because of our limited storage and compute, and we chose those particular variables because they were commonly used in previous works [1, 2, 3, 4, 5]. Using the above script, one can download any variable that is available in ERA5.
>
> > I feel resolution and variable set limitations are far more important than these at the moment.
>
> We wanted to emphasize that ClimateLearn is not limited by resolution or the variable set. One can always train their model on higher resolution data or include more variables using ClimateLearn if they have the compute to do so. The purpose of the experiments is not to achieve the best performance in the benchmark tasks, but to showcase the flexibility and functionality of ClimateLearn.
>
> > My main issue with the paper is that a large part of it is just reproducing other benchmarks and placing them all in a new repository without further modifications.
>
> We note that ClimateLearn is very different from simply placing other benchmarks in a commonplace. ClimateLearn is a single, unified package that allows easy benchmarking of deep learning models on various weather and climate problems. It is non-trivial and requires many modifications to provide a common interface across models, tasks, and datasets. Moreover, the paper introduces new results that were not studied in previous works, including benchmarking UNet and ViT on weather forecasting, climate downscaling, and climate projection, and studying extreme weather predictions and data robustness of deep learning models.
>
> > However, this paper sticks to the resolution of the old weatherbench paper/repo (which is quite coarse) and reproduces all their results. It is hard to justify benchmarking ML models at this coarse scale today
>
> Again, we wanted to emphasize that the only reason we benchmarked at this coarse resolution was the limited compute budget we have in an academic setting. Given this bottleneck, we focussed our experiments to showcase the breadth of possibilities with ClimateLearn library, one of which is performing weather forecasting. As we showed in the commands above, the library provides easy functionality to download and run the same set of experiments at finer resolutions, given sufficient compute and storage.
>
> > The data robustness to distributional shift is interesting and I have seen it in other papers as well w.r.t ERA5 and CMIP6 (including weatherbench). It would be useful to show another dataset from a climate model simulation. In Tab 2, what is the lead time?
>
> The experiment in WeatherBench only considered distributional shift from CMIP6 to ERA5 (training on CMIP6 and evaluating on ERA5) while we consider both directions. The lead time in Table 2 is 3 days. Since the CMIP6 data we used in Table 2 comes from a climate model simulation itself (MPI-HR), can the reviewer elaborate on why it would be useful to perform this distribution shift experiment using an additional dataset?

---

> > ### Author Response · Authors · 2023-08-22
> >
> > > Question: The authors mention PINNs which is an interesting new area of research in approximating the solution of PDEs with NNs. While I am not an NWP researcher, it is my understanding that weather/climate models contain many PDEs and parameterizations to close the system. Is the goal to include all these equations as a loss penalty as in PINNs?
> >
> > The goal of PINNs is to incorporate domain knowledge into the training of deep learning models. This can be done in multiple ways, e.g., by designing new neural network architectures that satisfy a hard physical constraint, or by adding penalty terms to encourage the model to satisfy a soft constraint.
> >
> > > How do the authors compare their work with modulus-launch from NVIDIA which aims to do the same thing (but also across other disciplines): training recipes, models and datasets for AI for science, and includes weather forecasting.
> >
> > Thanks for the pointer. The modulus-launch library seems to be in a relatively early stage, with the first version released 2 weeks back and many open issues. Modulus-launch focuses on providing training recipes for different neural network architectures, loss functions, and modeling choices on the weather forecasting task. In contrast, we support a breadth of datasets, metrics, and visualization functionalities across tasks above and beyond weather forecasting. Specifically, we currently support climate downscaling and climate projection tasks, CMIP6 and PRISM datasets, and corresponding metrics for these tasks, which are not supported by modulus-launch.
> >
> > > In the downscaling experiments, what is the ground truth for the high resolution variables? Do they just come from interpolated variables or is it from newer physics (such as pde solvers run on a higher resolution grid)?
> >
> > For ERA5 to ERA5 downscaling, we use the low and high resolution versions of the dataset. For ERA5 to PRISM downscaling, we used the PRISM dataset as the ground truth for the high-resolution variables. PRISM is not an interpolated version of ERA5, but an observational dataset actively maintained by the climate group at Oregon State University.
> >
> > [1] Rasp, Stephan, et al. "WeatherBench: a benchmark data set for data‐driven weather forecasting." Journal of Advances in Modeling Earth Systems 12.11 (2020): e2020MS002203.
> >
> > [2] Bi, Kaifeng, et al. "Pangu-weather: A 3d high-resolution model for fast and accurate global weather forecast." arXiv preprint arXiv:2211.02556 (2022).
> >
> > [3] Lam, Remi, et al. "GraphCast: Learning skillful medium-range global weather forecasting." arXiv preprint arXiv:2212.12794 (2022).
> >
> > [4] Nguyen, Tung, et al. "ClimaX: A foundation model for weather and climate." arXiv preprint arXiv:2301.10343 (2023).
> >
> > [5] Pathak, Jaideep, et al. "Fourcastnet: A global data-driven high-resolution weather model using adaptive fourier neural operators." arXiv preprint arXiv:2202.11214 (2022).

---

> > ### Comment · Reviewer_T3D2 · 2023-08-29
> >
> > Many thanks to the authors for responding to all my comments, I really appreciate it.
> > 1. Regarding the coarseness of the resolution, I completely understand that limited computational and storage budget limits your analysis and hence do support the acceptance of this paper. The authors mention that it can be easily extended for higher resolution variables, given sufficient compute/storage. I find this surprising. My understanding was that it is non-trivial to engineer ML dataloaders for high resolution datasets, model architectures that are scalable w.r.t to the input resolution and I'm unsure that this package can be easily extended for these. For example, the authors mention benchmarking ViT. I do not think a large ViT could even process a high resolution image tensor with 100s of channels without overflowing the GPU memory (since activations take up a lot of memory here), which would necessitate model parallelism techniques etc: all these are significantly non-trivial; and not to mention I/O bottlenecks when reading such high volume data. Hence, I'm not sure if the response completely addressed my concerns here.
> > 2. Another issue with benchmarking at coarse resolution is: do lessons learnt at coarse resolution transfer to finer resolutions? I would not think that this is also a trivial issue.
> > 3. Regarding distribution shifts, there are many climate models run with different boundary conditions regarding CO2 concentrations. I was just wondering what the effect of transfer learning is based on the underlying distribution shift (as quantified by the above boundary conditions). One would expect a model simulated with extreme increase in temperatures would have a harder time transfering to ERA5 or vice-versa. However, this was just a comment and a little unrelated to the contributions of the paper, so I don't think this warrants any more experiments right now.
> >
> > I do recognize the utility of the package in its current form as a way for ML+weather researchers to get bootstrapped into this line and hence think the paper can be accepted. However, I retain my score of 6 (weak accept).

---

### Official Review · Reviewer_ofXC · 2023-07-21
**Software package with unclear contribution**

**Rating:** 6
**Confidence:** 4
**Clarity:** Clarity is fine.

**Strengths:**

1. The library unifies handling a number of different weather datasets, which can be challenging for non-experts to work with.
2. Providing model implementations is also helpful for practitioners to quickly get started.

**Additional Feedback:**

A couple of minor comments:
- L28: I believe a GCM is simply a kind of numerical weather prediction model; why make a distinction?
- L34: The paper states that numerical models cannot utilize observational data, which seems wrong to me. The data assimilation process definitely incorporates observations from weather stations, satellites, and other sources.
- L36: The paper mentions petabyte-scale climate data in the introduction. While this is definitely true (indeed, ECMWF adds about 200 TB of data per day to their archive), ClimateLearn does not seem like it is particularly helpful with that; managing such large data volumes requires highly specialized and performant data management systems, none of which are discussed here.

**Correctness:**

No correctness issues in general. However, the experimental results do not include error bars.

**Documentation:**

Documentation is adequate and includes code for the experiments as well as examples.

**Ethics:**

No ethical concerns.

**Limitations:**

Limitations are discussed.

**Opportunities For Improvement:**

1. In terms of unifying dataset access, how does this differ from existing interfaces to weather/climate datasets, such as CliMetLab (https://climetlab.readthedocs.io/en/latest/)?
2. The existing tasks that ClimateLearn supports are quite limited. How hard is it to add additional tasks, such as probabilistic forecasting or working with ensembles?
3. The paper appears to introduce a new subset of ERA5, "Extreme-ERA5", for extreme weather forecasting. While this is quite valuable, this seems like it is not well-documented here and may be more suitable as a separate paper. It is also unclear whether the approach to constructing thresholds to identify extreme events is suitable without further evaluation. It may be better to instead utilize standard metrics for this, such as the Extreme Forecast Index. (For an example of using the EFI in ML, see Ashkboos et al., "ENS-10: A Dataset For Post-Processing Ensemble Weather Forecasts", NeurIPS 2022.)
4. The supported traditional baselines seem quite simple (although they are important standard baselines). It would be better to include more challenging baselines; for example, WeatherBench includes predictions from the operational IFS model as a very strong baseline. It may also be worth considering SotA deep learning models, such as FourCastNet or MeshGraphNet for forecasting tasks.
5. What does the comparison of direct, continuous, and iterative forecasting in Section 4.1.1 add over existing works? This has been studied in other contexts (e.g., in the original WeatherBench paper [ref 56] and in [ref 55], among others).

Overall, it is not clear to me what ClimateLearn adds over existing frameworks, and the experiments seem to primarily be repeating what has already been done in the literature (e.g., in WeatherBench and its various submissions).

-----

The authors have addressed a number of my above concerns and I have raised my score.

**Relation To Prior Work:**

Some relevant prior work in weather benchmarking frameworks is not discussed (see above).

**Summary And Contributions:**

The paper introduces ClimateLearn, a Python library to manage different weather datasets and support machine learning applications using them. It also includes various metrics and model implementations. Results on weather forecasting and climate downscaling are presented.

---

> ### Author Response · Authors · 2023-08-22
>
> We thank the reviewer for the constructive feedback and for the recognition of the helpfulness and contributions of ClimateLearn. We have updated the paper to show more experiments and new features of the package, as well as to clarify certain points in the paper. We answer each of the reviewer's concerns below.
>
> > In terms of unifying dataset access, how does this differ from existing interfaces to weather/climate datasets, such as CliMetLab
>
> CliMetLab and ClimateLearn share similarities in terms of providing easy access to weather and climate data, since both libraries aim to simplify the process of downloading, preprocessing, and making climate data ML-friendly.
> We also wanted to note that while CliMetLab focuses purely on providing easy access to climate data, ClimateLearn is an end-to-end library for training and evaluating machine learning models on weather and climate problems. Specifically, ClimateLearn implements multiple predefined tasks and metrics, simple baselines and deep learning models, and visualizations, which are not provided by CliMetLab.
> One practical difference between the two is while CliMetLab is intended to be used in Jupyter notebooks (quoting their readme “it is mostly intended to be used in Jupyter notebooks”), ClimateLearn encompasses a wider range of usecases, from running quick starter code in Jupyter (https://colab.research.google.com/drive/1LcecQLgLtwaHOwbvJAxw9UjCxfM0RMrX?usp=sharing) to large-scale benchmarking experiments as we showed in the paper. As such, researchers and practitioners can directly utilize our library to develop and benchmark against high-quality deep learning baselines, which we do not believe is straightforward with CliMetLab. We have included this discussion in the updated version of the paper.
>
> > The existing tasks that ClimateLearn supports are quite limited. How hard is it to add additional tasks, such as probabilistic forecasting or working with ensembles?
>
> ClimateLearn currently supports weather forecasting, climate projection, and climate downscaling, three major tasks that were considered in many previous works [1, 2, 3, 4, 5, 6, 7, 8, 9, 10]. To the best of our knowledge, ClimateLearn supports the most number of weather and climate tasks among similar efforts. Moreover, ClimateLearn is built in a way that allows easy extension to new models, tasks, and datasets, so one can always extend the library to add their task of interest. Probabilistic forecasting is straightforward to include and would involve two changes: (a) simple architecture modifications, e.g., probabilistic outputs, or aggregating multiple training runs for ensembling; (b) new evaluation metrics, e.g., CRPS and Spread-skill ratio. Both these changes are compatible with our API design. To demonstrate this, we added probabilistic forecasting to the suit of tasks, and included a new experiment in Table 4 (Section 5.2) in the updated paper which benchmarks two models based on ResNet, one in which is an ensemble model and the other is a parametric model.
>
> > The paper appears to introduce a new subset of ERA5, "Extreme-ERA5", for extreme weather forecasting. While this is quite valuable, this seems like it is not well-documented here and may be more suitable as a separate paper.
>
> Since Extreme-ERA5 is not a completely new dataset but a variant of ERA5, and as it’s already included in the library, we believe it would be better to include it in the ClimateLearn paper. We documented in detail how to construct Extreme-ERA5 from the original ERA5 in Section 3.2 and Appendix B.2.2. Can the reviewer elaborate on what detail is missing?
>
> > It is also unclear whether the approach to constructing thresholds to identify extreme events is suitable without further evaluation. It may be better to instead utilize standard metrics for this, such as the Extreme Forecast Index.
>
> Indeed, we note that the definition of extreme weather events is an active area of debate and research within the research community [11]. We picked a relatively simple and interpretable definition to provide insights into how well different deep learning models predict extreme values. This was also considered in previous work [12].
>
> Thanks for the suggestion on the Extreme Forecast Index! It seems the index applies to deployment scenarios where we want to compare the forecast distribution with the model climate (under the assumption that the model climate is an accurate reflection of the ground-truth climate). We will incorporate these metrics as well in our library in consultation with domain experts to help define some of the design choices (e.g, choice of members).

---

> > ### Author Response · Authors · 2023-08-22
> >
> > > It would be better to include more challenging baselines; for example, WeatherBench includes predictions from the operational IFS model as a very strong baseline. It may also be worth considering SotA deep learning models, such as FourCastNet or MeshGraphNet for forecasting tasks.
> >
> > We already included the operational IFS model in Figure 2 in the paper as a strong baseline for the weather forecasting task. In addition, we added ClimaX and FourCastNet to ClimateLearn and benchmarked these two methods in an additional experiment in Figure 4 (Section 5.1) in the updated version of the paper. We’ll keep adding new models to the ClimateLearn that can be benchmarked using the same pipeline/implementation in the library.
> >
> > > What does the comparison of direct, continuous, and iterative forecasting in Section 4.1.1 add over existing works? This has been studied in other contexts (e.g., in the original WeatherBench paper [ref 56] and in [ref 55], among others).
> >
> > We perform this experiment to demonstrate the functionalities of ClimateLearn in supporting different modeling choices so that users have an easy way to trade off performance and efficiency when training their own model. While WeatherBench already showed a similar result, it did not provide an easily accessible implementation. ClimateLearn also provides support for more modern architectures such as ViTs and UNets which were not considered in WeatherBench and are increasingly being considered in recent papers. Moreover, we evaluate the performance on more lead times than WeatherBench and show the out-of-disitribution lead time performance of the continuous model which was not studied in WeatherBench.
> >
> > > Overall, it is not clear to me what ClimateLearn adds over existing frameworks, and the experiments seem to primarily be repeating what has already been done in the literature (e.g., in WeatherBench and its various submissions).
> >
> > We emphasize that ClimateLearn is very different from existing frameworks, and believe that it should not be viewed simply as a benchmark proposing a new dataset/model (e.g., WeatherBench). ClimateLearn is a ML software platform that allows easy end-to-end training and evaluation of different ML models on a broad range of tasks in climate science with various qualitative and quantitative metrics. In contrast, previous works supply only individual components of an ML pipeline, which we refer to the Related work for a detailed discussion.
> >
> > Specific to the reviewer’s question, ClimateLearn extends WeatherBench on many axes. First, ClimateLearn provides support for climate downscaling and climate projection, which were not considered in WeatherBench. Second, ClimateLearn implements a variety of deep learning models and studies their performance in extreme weather prediction and transferability between different datasets while WeatherBench does not. Third, ClimateLearn supports visualization of the prediction of deep learning models. More generally, ClimateLearn is developed with a general-purpose API that permits easy extensibility to support new datasets, models, metrics, tasks while preserving its modular and reusable abstraction – a functionality which does not have a direct analog with any individual benchmark such as WeatherBench. In terms of broader impacts, ClimateLearn provides an easy-to-use and well documented Pytorch library that lowers the barrier for ML researchers and climate scientists who want to use ML to solve climate problems, which we believe is a significant contribution to research and education in this domain.

---

> > > ### Author Response · Authors · 2023-08-22
> > >
> > > > Minor comments
> > >
> > > We thank the reviewer for pointing these out.
> > > - L28: We agree with the reviewer. We have updated this in the revised paper to avoid confusion.
> > > - L34: We meant to say that the process of incorporating observational data into numerical models is laborious whereas ML models allow doing so in a more automated way. We have updated the paper to clarify this point.
> > > - L36: We mentioned petabyte-scale climate data in the paper only to motivate the development of ML models as a promising alternative to numerical models.
> > >
> > > [1] Rasp, Stephan, et al. "WeatherBench: a benchmark data set for data‐driven weather forecasting." Journal of Advances in Modeling Earth Systems 12.11 (2020): e2020MS002203.
> > >
> > > [2] Bi, Kaifeng, et al. "Pangu-weather: A 3d high-resolution model for fast and accurate global weather forecast." arXiv preprint arXiv:2211.02556 (2022).
> > >
> > > [3] Lam, Remi, et al. "GraphCast: Learning skillful medium-range global weather forecasting." arXiv preprint arXiv:2212.12794 (2022).
> > >
> > > [4] Nguyen, Tung, et al. "ClimaX: A foundation model for weather and climate." arXiv preprint arXiv:2301.10343 (2023).
> > >
> > > [5] Pathak, Jaideep, et al. "Fourcastnet: A global data-driven high-resolution weather model using adaptive fourier neural operators." arXiv preprint arXiv:2202.11214 (2022).
> > >
> > > [6] Baño-Medina, Jorge, Rodrigo Manzanas, and José Manuel Gutiérrez. "Configuration and intercomparison of deep learning neural models for statistical downscaling." Geoscientific Model Development 13.4 (2020): 2109-2124.
> > >
> > > [7] Liu, Yumin, Auroop R. Ganguly, and Jennifer Dy. "Climate downscaling using YNet: A deep convolutional network with skip connections and fusion." Proceedings of the 26th ACM SIGKDD International Conference on Knowledge Discovery & Data Mining. 2020.
> > >
> > > [8] Rodrigues, Eduardo Rocha, et al. "DeepDownscale: A deep learning strategy for high-resolution weather forecast." 2018 IEEE 14th International Conference on e-Science (e-Science). IEEE, 2018.
> > >
> > > [9] Sachindra, D. A., et al. "Statistical downscaling of precipitation using machine learning techniques." Atmospheric research 212 (2018): 240-258.
> > >
> > > [10] Watson‐Parris, Duncan, et al. "ClimateBench v1. 0: A Benchmark for Data‐Driven Climate Projections." Journal of Advances in Modeling Earth Systems 14.10 (2022): e2021MS002954.
> > >
> > > [11] Julien Cattiaux and Aurélien Ribes.  Defining Single Extreme Weather Events in a Climate Perspective. Bulletin of the American Meteorological Society. 2018.
> > >
> > > [12] Blanusa, Mackenzie L., Carla J. López-Zurita, and Stephan Rasp. "Internal variability plays a dominant role in global climate projections of temperature and precipitation extremes." Climate Dynamics (2023): 1-15.

---

> > > > ### Comment · Reviewer_ofXC · 2023-08-24
> > > > **Response**
> > > >
> > > > Thank you for the detailed response, clarifications, and updates to the paper. They've addressed a number of my concerns and I've raised my score accordingly.
> > > >
> > > > > Specific to the reviewer’s question, ClimateLearn extends WeatherBench on many axes.
> > > >
> > > > I agree that ClimateLearn seems to have a number of software advantages over WeatherBench, but I think it's important to note that, unlike WeatherBench, ClimateLearn does not seem to be a _benchmark_ (with an associated community leaderboard and so on), but rather a tool one might use to define a benchmark.

---

### Official Review · Reviewer_48aK · 2023-07-24
**delayed review of ClimateLearn dataset paper**

**Rating:** 8
**Confidence:** 4

**Strengths:**

The paper addresses a hot topic in weather and climate research, i.e. how we can develop new machine learning models that might make forecasting and modelling tasks faster, and possibly, even better than established forecasting tools. This is certainly a rapidly growing research area so that the contribution made by this paper is an important step in the direction of comparability and robustness. It could also greatly facilitate the inclusion of pure machine learners in developing these tools, and similarly open up new routes for scientists to provide their input into a common modelling pipeline.

The paper is mostly well-written and well-structured. It was a pleasure to read and uses clear language throughout. Clearly, the topic relates to questions of grand social and ethical importance.

**Additional Feedback:**

@AreaChairs: I meant to send you a message about my delayed review, but following your email instructions I found no way to get in touch without completing the entire review anyway. I have got two more papers to review. Could you let me know if you don't want me to submit those due to the delay? I don't want to mention those papers here for confidentiality.

**Clarity:**

Mostly yes. Suggestions for minor improvements are:
- l. 104-106: shape of input lacks clarity. H and W denote the spatial coverage, but does this include altitude/pressure levels, or are these considered as separate variables?
- l. 108: 'at global scale' -> does this mean global mean or all grid points/pixels globally? Best to clarify.
- l. 131: 400,000 data points? For each timestep?
- l. 165: what is meant by 'climatology' here? For daily and monthly data, for example, I would expect that a climatology includes a seasonal cycle, but from your text it is not clear to me if this just means a longer term (annual) average value, which would be a really bad baseline that could be easily improved upon.
- l. 183: similar - Anomaly CC -> relative to what?


**Correctness:**

Comments:
- Figure 1: Typo in lower right corner 'Uncertainity'
- Typo line 79: precipatation
- l. 228: and standard deviation 1 (change order of words for clarity?) Same in line 306.
- l. 241: works (plural) with just one citation? I feel this should be singular 'work'
- Table 2: the large RMSEs in the first row seem very odd? Different units in different datasets? Otherwise, how is this explained? I really can't make sense of this and suspect an error.


**Documentation:**

I didn't have time to check every detail, but the big picture looks fine to me. Of course, I would encourage the authors to re-check all software and data contributions before submission for completeness and functionality.

**Ethics:**

No.

**Limitations:**

My main concern wrt limitations is the differentiation between the weather and climate part of the contribution and how these differ. Long-term weather records will of course include a climate change signal, and I expect those to differ between ERA5 and CMIP6. Therefore, a few results were surprising, but this hasn't really been reflected on by the authors, see comment:

- l. 295-297: could you hypothesize why training on CMIP6's longer dataset always performs better? This appears to be a battle between sample size and the models' vs the true climate sensitivity...this is just based on the MPI-HR model?

**Opportunities For Improvement:**

Main comment: I found the name 'ClimateLearn' misleading and, accordingly, had imagined an entirely different focus of the paper. While it is useful to bundle climate and weather datasets together, the focus of the paper, especially of the main part, is on weather forecasting and downscaling. The introduction, dataset name, and conclusions instead almost exclusively talk about the challenge of climate change. I find this hard to square in my mind and would suggest renaming the dataset according to its focus, moving it at least somewhat closer to 'WeatherBench'. The climate contribution is also the weakest part in my opinion, building entirely on the existing ClimateBench, overselling the novelty of the contribution made in this domain, and being only covered in the Supplementary. The discussion of this dataset also appears to reveal certain weaknesses in the authors' understanding of climate change modelling and what machine learning can do in this extrapolation setting (the climate is non-stationary...), see also detailed comments below.

Further comments:
- l. 95: 'climate projection'? This sounds as if the machine learning model can project climate change, but what it actually does is to emulate a physics-based climate model that does this job. Therefore, 'climate model emulation' would be the more correct term. Same in line 118.
- l. 134-142: For Extreme-ERA5 you calculate percentiles. However, given the underlying warming trend these percentiles will be shifting. How is this considered? How will this be updated in future versions of the dataset?
- l. 295-297: could you hypothesize why training on CMIP6's longer dataset always performs better? This appears to be a battle between sample size and the models' vs the true climate sensitivity...this is just based on the MPI-HR model?
- Acknowledgements need to officially acknowledge the World Climate Research Programme (WCRP) for CMIP6 as well.

**Relation To Prior Work:**

Overall yes.

**Summary And Contributions:**

The paper introduces a PyTorch library 'climate-learn' that bundles existing datasets for (a) weather and its extremes (using ERA5 data), (b) downscaling (ERA5 and PRISM) and (c) climate model emulation (using ClimateBench, which in turn is derived from the NorESM model in CMIP6). The paper further provides code to train (very basic) baselines on these datasets for specific tasks and compares their performance to more complex deep learning architectures. In addition, it compares the results to simple standard metrics in the form of e.g. persistence and a climatology (where relevant). The paper certainly makes a valuable contribution to facilitate and standardize machine learning intercomparisons in this rapdily growing research area.

---

> ### Author Response · Authors · 2023-08-22
>
> We thank the reviewer for the constructive feedback and for the recognition of the significance of ClimateLearn and the good presentation of the paper. We have updated the paper to show more experiments and new features of the package, as well as to clarify certain points in the paper. We answer each of the reviewer's concerns below.
>
> > I found the name 'ClimateLearn' misleading and, accordingly, had imagined an entirely different focus of the paper.
>
> Thanks for the suggestion. We wanted to have a name that conveys the idea of using a common set of building blocks for machine learning to solve both weather and climate problems, since we plan to expand the suit of tasks to both regimes in the future. In that regard, we thought of ClimateLearn that appeals to the growing community at the intersection of AI and Climate Change which studies both weather and climate problems. How about AtmosphereLearn? If the reviewer is fine with it, we are open to considering the name change in the final version.
>
> > The climate contribution is also the weakest part in my opinion, building entirely on the existing ClimateBench, overselling the novelty of the contribution made in this domain, and being only covered in the Supplementary.
>
> We believe the climate projection task is important to study, and we only show it in the Appendix because of the limited space. While we used the dataset and metrics from ClimateBench, we benchmarked other deep learning architectures that were not considered in ClimateBench, as well as made engineering contributions in the library integrating it within the same API as our other tasks (forecasting, downscaling).
>
> > l. 95: 'climate projection'?
>
> This is taken directly from the ClimateBench, we did not give it this name.
>
> > l. 134-142: For Extreme-ERA5 you calculate percentiles. However, given the underlying warming trend these percentiles will be shifting. How is this considered? How will this be updated in future versions of the dataset?
>
> The percentiles are calculated using the training data, similarly to previous work [1]. As we observe more data points, we can expand the training period and create new releases of extreme ERA5 on a regular basis so that extremes are updated over time as the climate changes.
>
> > l. 295-297: could you hypothesize why training on CMIP6's longer dataset always performs better? This appears to be a battle between sample size and the models' vs the true climate sensitivity...this is just based on the MPI-HR model?
>
> Yes, this is based on the MPI-HR model, and we also hypothesize that better performance comes from having a longer training period.
>
> > Acknowledgements need to officially acknowledge the World Climate Research Programme (WCRP) for CMIP6 as well.
>
> We updated the paper to include this.
>
> > Correctness
>
> Thank you for pointing these out. We fixed the typos and improved clarity according to the reviewer’s suggestions.
>
> > Table 2: the large RMSEs in the first row seem very odd? Different units in different datasets? Otherwise, how is this explained? I really can't make sense of this and suspect an error.
>
> This error comes from different units of geopotential in ERA5 and CMIP6. We fixed this in the updated version.
>
> > Clarity
>
> Thank you for pointing these out. We updated the paper to clarify these points. The climatology we used in our original submission was computed over all time steps in the training dataset, which was also considered in WeatherBench [2]. The Anomaly CC is relative to this climatology. We additionally implemented a weekly climatology baseline, which includes a mean computed for each of the 52 calendar weeks. We updated the paper to include this baseline in Figure 2.
>
> [1] Blanusa, Mackenzie L., Carla J. López-Zurita, and Stephan Rasp. "Internal variability plays a dominant role in global climate projections of temperature and precipitation extremes." Climate Dynamics (2023): 1-15.
>
> [2] Rasp, Stephan, et al. "WeatherBench: a benchmark data set for data‐driven weather forecasting." Journal of Advances in Modeling Earth Systems 12.11 (2020): e2020MS002203.

---

> > ### Comment · Reviewer_48aK · 2023-08-29
> > **AtmosphereLearn**
> >
> > Thank you for addressing my comments. I like AtmosphereLearn as title and it would better reflect the balance between weather and climate, especially given the current set-up.

---

### Official Review · Reviewer_9UcC · 2023-07-28
**Useful contributions, simple baselines**

**Rating:** 7
**Confidence:** 4
**Correctness:** So far as I can tell it is correct.
**Clarity:** Yes, it is very well-written.

**Strengths:**

It's highly valuable and timely for the climate informatics community to have a readily accessible set of tools for accessing a variety of climate data and benchmarking ML algorithms on it. The package is clearly still in development and missing various functionalities, but even in this form it should be quite useful.

**Additional Feedback:**

N/A

**Documentation:**

Yes, there is sufficient detail.

**Ethics:**

No ethics issues.

**Limitations:**

No additional limitations need to be discussed.

**Opportunities For Improvement:**

The baselines presented in the work are simple and do not include SoTA forecasting methods such as FourCastNet, ClimaX, Pangu-Weather, etc. For downscaling, superresolution GANs are not included. This limits the utility of the dataset in its current form, but is not a fatal flaw.

**Relation To Prior Work:**

Related work is appropriately discussed.

**Summary And Contributions:**

The authors provide a framework for downloading climate / weather data and testing ML models on common applications in the domain (forecasting / downscaling). They also provide a suite of simple ML models (though not more complicated SoTA models) for testing, and analyze the results of applying these models.

---

> ### Author Response · Authors · 2023-08-22
>
> We thank the reviewer for the constructive feedback and for the recognition of the usefulness and significance of ClimateLearn. We have updated the paper to show more experiments and new features of the package, as well as to clarify certain points in the paper. We answer each of the reviewer's concerns below.
>
> > The baselines presented in the work are simple and do not include SoTA forecasting methods such as FourCastNet, ClimaX, Pangu-Weather, etc. For downscaling, superresolution GANs are not included. This limits the utility of the dataset in its current form, but is not a fatal flaw.
>
> Thanks for the suggestion. In our first version of this platform, our focus was on developing ClimateLearn as a long-term sustainable project that allows easy extension to new models, tasks, and datasets by us and others (via open-source). During the rebuttal, we added ClimaX and FourCastNet to ClimateLearn, and included their benchmarking in Figure 4 (Section 5.1) in the updated version of the paper.
> As the library’s usage expands, we will implement more complex models in future versions, and hope to benefit from open-source contributions as well.

---

> > ### Comment · Reviewer_9UcC · 2023-08-29
> > **Thank you for the responses**
> >
> > Thank you to the authors for these responses. I continue to believe this submission represents a strong contribution and should be accepted.

---

### Official Review · Reviewer_qd4b · 2023-07-28
**Ambitious motivation but the execution is oversimplified**

**Rating:** 6
**Confidence:** 3
**Correctness:** yes
**Clarity:** very much so

**Strengths:**

The manuscript is well-written and easy to understand which makes me feel hopeful that utilization by non-experts (an ongoing problem with these specific tasks in the field) may be straightforward. This package seems like it would be useful to students, in particular.

**Additional Feedback:**

n/a

**Documentation:**

yes in supplement

**Limitations:**

Limitations are discussed but not in enough detail.

**Opportunities For Improvement:**

As a domain scientist, I can't help but be uncomfortable about some of the over-simplification of some things (e.g., defining extremes using simplified percentiles and defining climatology as a seasonally-invariant mean) that seem to be made in order to make it easier to package everything in an easy-to-understand way and to join the efforts along their shared axes. It is hard to imagine that this contribution will help advance the frontier. I appreciate the desire for unification due to some similarity in the data, but there is a reason why the tasks of weather forecasting, climate prediction, and downscaling have remained very much separate both in the non-ML domain and in machine learning efforts to address them: they are pretty different.

**Relation To Prior Work:**

yes

**Summary And Contributions:**

This contribution is an attempt to construct a unified and encompassing benchmark library for machine learning efforts in atmospheric science, specifically for (1) weather forecasting (2) climate prediction, and (3) downscaling. Currently, datasets and libraries for these three tasks exist separately despite similarity in the data. The primary contribution of this work is bringing together a few datasets, and constructing a library of existing models, evaluation metrics, and visualization code to make this work accessible.

---

> ### Author Response · Authors · 2023-08-22
>
> We thank the reviewer for the constructive feedback and for the recognition of the usefulness of ClimateLearn and the good presentation of the paper. We have updated the paper to show more experiments and new features of the package, as well as to clarify certain points in the paper. We answer each of the reviewer's concerns below.
>
> > Over-simplification of some things (e.g., defining extremes using simplified percentiles and defining climatology as a seasonally-invariant mean) that seem to be made in order to make it easier to package everything in an easy-to-understand way and to join the efforts along their shared axes.
>
> These were only design choices we made for the experiments in the paper and not intended to be general-purpose recommendations. We added a weekly climatology baseline [1]  in which we compute a mean for each of the 52 calendar weeks, and included this baseline in Figure 2 in the updated version of the paper.
>
> ERA5-Extreme: Previous work [9] also used percentiles to determine weather extremes. We believe while simple, this dataset still provides valuable insights into how well different deep learning models predict extreme values. Moreover, the definition of extreme weather is an open research direction even for the community with no single standard definition. In the future, we will release new versions of Extreme-ERA5 which utilize other metrics.
>
> > I appreciate the desire for unification due to some similarity in the data, but there is a reason why the tasks of weather forecasting, climate prediction, and downscaling have remained very much separate both in the non-ML domain and in machine learning efforts to address them: they are pretty different.
>
> We agree that the different tasks have quite distinct nuances. However, we note that our library does not advocate for a uniform approach for any of these tasks, but leaves that choice entirely up to the practitioner. What the library does is provide a common software interface for end-to-end benchmarking and in that respect, many general components are shared across tasks, e.g., datasets such as ERA5 [2, 3, 4, 5, 6, 7] and CMIP6 [5, 7, 8], metrics such as (latitude weighted) RMSE [1, 4, 8], and architectures such as CNNs [1, 4, 8] and ViTs [2, 5, 6], have been used across different tasks.
>
> [1] Rasp, Stephan, et al. "WeatherBench: a benchmark data set for data‐driven weather forecasting." Journal of Advances in Modeling Earth Systems 12.11 (2020): e2020MS002203.
>
> [2] Bi, Kaifeng, et al. "Pangu-weather: A 3d high-resolution model for fast and accurate global weather forecast." arXiv preprint arXiv:2211.02556 (2022).
>
> [3] Lam, Remi, et al. "GraphCast: Learning skillful medium-range global weather forecasting." arXiv preprint arXiv:2212.12794 (2022).
>
> [4] Nagasato, Takeyoshi, et al. "Extension of Convolutional Neural Network along Temporal and Vertical Directions for Precipitation Downscaling." arXiv preprint arXiv:2112.06571 (2021).
>
> [5] Nguyen, Tung, et al. "ClimaX: A foundation model for weather and climate." arXiv preprint arXiv:2301.10343 (2023).
>
> [6] Pathak, Jaideep, et al. "Fourcastnet: A global data-driven high-resolution weather model using adaptive fourier neural operators." arXiv preprint arXiv:2202.11214 (2022).
>
> [7] Rasp, Stephan, and Nils Thuerey. "Data‐driven medium‐range weather prediction with a resnet pretrained on climate simulations: A new model for weatherbench." Journal of Advances in Modeling Earth Systems 13.2 (2021): e2020MS002405.
>
> [8] Watson‐Parris, Duncan, et al. "ClimateBench v1. 0: A Benchmark for Data‐Driven Climate Projections." Journal of Advances in Modeling Earth Systems 14.10 (2022): e2021MS002954.
>
> [9] Blanusa, Mackenzie L., Carla J. López-Zurita, and Stephan Rasp. "Internal variability plays a dominant role in global climate projections of temperature and precipitation extremes." Climate Dynamics (2023): 1-15.

---

### Decision · Program_Chairs · 2023-09-22

**Decision:**

Accept (Poster)

**Comment:**

ClimateLearn (to be renamed AtmosphereLearn, as per the reviews) is a packaging of previous methods and datasets focused on the atmospheric sciences. The focus on this unique and important use-case adds considerable novelty, especially as data in this domain is not easily understood by newcomers without the appropriate domain expertise. After the rebuttal period, all reviewers unanimously agreed that the work should be accepted. Of note, their package makes it easy to work with either low-resolution versions of the data (which are easy to use with academic-scale compute) or very high resolution versions of the data for the development of SOTA methods. As such this package will have broad utility in the field. Reviewers note that this particular contribution itself is not a benchmark, but rather a tool one could use to define a benchmark. In light of the novelty of the contribution on the data side and the ease of use, reviewers still believed this work to be of considerable interest to the community, despite the lack of a proper benchmark. I believe this speaks to the strength of the dataset contribution. This package is also intended to grow and be developed by what remains a niche community, so this work represents and important tool for facilitating access to the hard problem of using ML in the atmospheric sciences.